# Molecular mechanism on forcible ejection of ATPase inhibitory factor 1 from mitochondrial ATP synthase

Ryohei Kobayashi [1,2], Hiroshi Ueno[1], Kei-ichi Okazaki [2,3] & Hiroyuki Noji [1] ✉

$IF_1$ is a natural inhibitor protein for mitochondrial $F_oF_1$ ATP synthase that blocks catalysis and rotation of the $F_1$ by deeply inserting its N-terminal helices into $F_1$. A unique feature of $IF_1$ is condition-dependent inhibition; although $IF_1$ inhibits ATP hydrolysis by $F_1$, $IF_1$ inhibition is relieved under ATP synthesis conditions. To elucidate this condition-dependent inhibition mechanism, we have performed single-molecule manipulation experiments on $IF_1$-inhibited *bovine* mitochondrial $F_1$ ($bMF_1$). The results show that $IF_1$-inhibited $F_1$ is efficiently activated only when $F_1$ is rotated in the clockwise (ATP synthesis) direction, but not in the counterclockwise direction. The observed rotational-direction-dependent activation explains the condition-dependent mechanism of $IF_1$ inhibition. Investigation of mutant $IF_1$ with N-terminal truncations shows that the interaction with the γ subunit at the N-terminal regions is crucial for rotational-direction-dependent ejection, and the middle long helix is responsible for the inhibition of $F_1$.

$F_oF_1$-ATP synthase ($F_oF_1$) is a rotary motor protein that catalyzes ATP synthesis reaction from ADP and inorganic phosphate ($P_i$) using the proton motive force (*pmf*) across membranes. $F_oF_1$ is composed of two rotary motor proteins: $F_o$ and $F_1$[1–5]. $F_o$ is a membrane-embedded protein complex forming the proton translocation pathway. When $F_oF_1$ synthesizes ATP, $F_o$ transports protons from the outside to the inside of the membrane, accompanied by the clockwise rotation of the rotor complex inside $F_oF_1$ when viewed from the outside of the membrane. $F_1$ is a water-soluble portion of $F_oF_1$ that contains catalytic centers for ATP synthesis. Proton translocation by $F_o$ and ATP synthesis/hydrolysis reactions in $F_1$ are tightly coupled through the rotation of the rotor complex. When *pmf* is sufficiently high, $F_o$ forcibly rotates $F_1$, powered by proton translocation down *pmf*, resulting in ATP synthesis via $F_1$. When *pmf* is insufficient, $F_1$ reverses the rotation and hydrolyzes ATP, which induces $F_o$ to pump protons to generate *pmf*[6].

$F_1$ is an ATPase that hydrolyzes ATP to rotate its rotor part counterclockwise, when viewed from the membrane side, against the surrounding stator $\alpha_3\beta_3$-ring[7], where the central rotor γ subunit is accommodated in the central cavity[8–12]. $F_1$ possesses three catalytic

sites for ATP hydrolysis/synthesis, each at the interface between the α and β subunits. Amino acid residues critical for catalysis are mainly located in the β subunit. The "ground-state" crystal structure of $bMF_1$[13] revealed that the three β subunits have different conformations and nucleotide-bound states; β with nucleotide analog ($\beta_{TP}$), β with ADP ($\beta_{DP}$), and β with none ($\beta_E$). The $\beta_{TP}$ and $\beta_{DP}$ adopt a closed form with their C-terminal domain swinging towards the γ subunit, whereas $\beta_E$ adopts an open conformation[4,12,14]. The rotary dynamics of $F_1$ have been investigated extensively using single-molecule studies[1,15–19]. The reaction scheme of rotary catalysis in $bMF_1$[20] (Fig. 1a) was proposed by considering the structural features of $bMF_1$ and kinetic analysis in single-molecule studies on $F_1$ from thermophilic *Bacillus* PS3 ($TF_1$)[21]. Several experiments have shown that most of the crystal structures of $bMF_1$ represent catalytic dwell states[22–24].

Most $F_oF_1$ play the primary role in ATP synthesis in vivo, although proton- or sodium-pumping activity driven by ATP hydrolysis is dominant in some species. In general, ATP hydrolysis activity of $F_1$ can result in futile ATP consumption. Therefore, several types of regulatory systems can suppress or block the ATP hydrolysis activity

[1]Department of Applied Chemistry, Graduate School of Engineering, The University of Tokyo, Bunkyo-ku, Tokyo 113-8656, Japan. [2]Research Center for Computational Science, Institute for Molecular Science, Okazaki, Aichi 444-8585, Japan. [3]The Graduate University for Advanced Studies, SOKENDAI, Okazaki, Aichi 444-8585, Japan. ✉e-mail: hnoji@g.ecc.u-tokyo.ac.jp

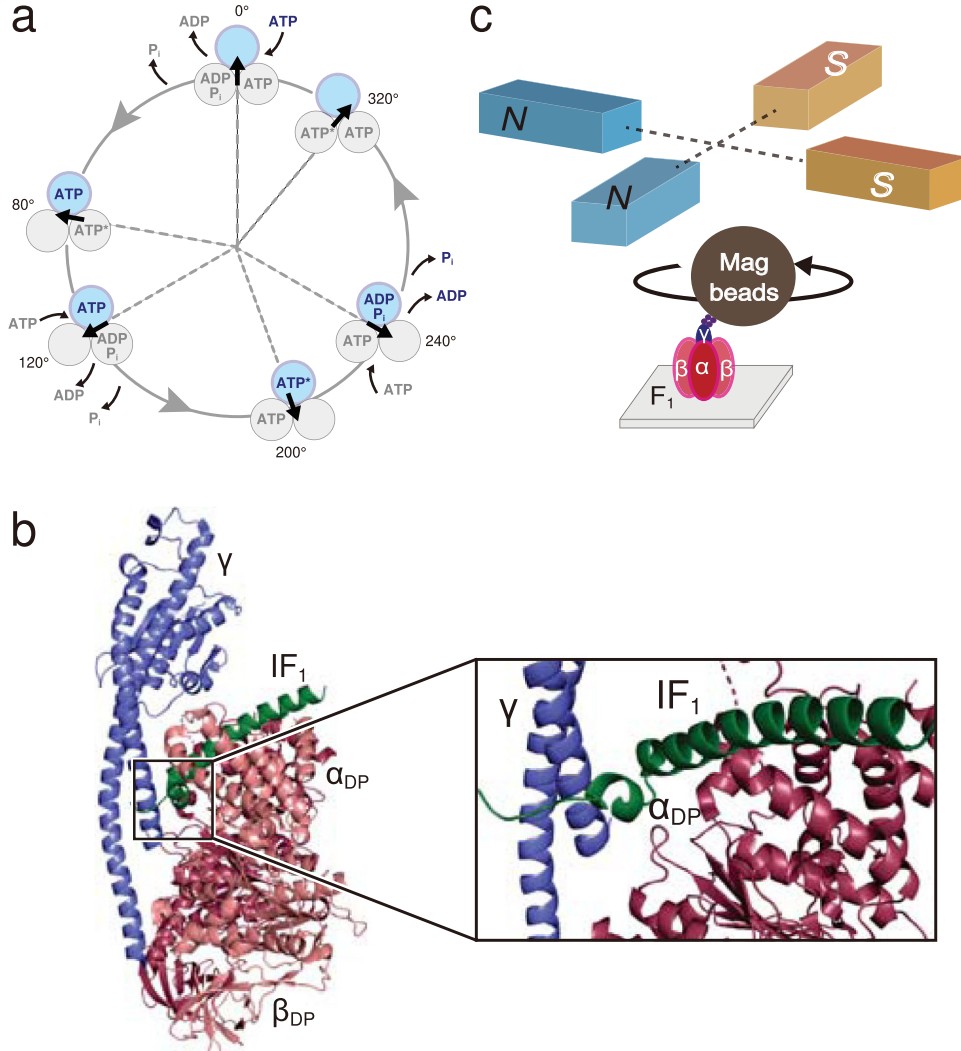

**Fig. 1 | Outline of this work. a** Rotation scheme of $b$MF$_1$. The circles represent the catalytic states of bound nucleotides at the β subunit. ATP* in the circles at 80°, 200°, and 320° represents the catalytically active state in which the bound ATP is to be hydrolyzed. The arrows represent the rotary angles of γ subunit. 0° is defined as the position of the γ subunit where the blue β subunit binds to ATP. *Short pauses* at 10°-20°, 130°-140°, and 250°-260°, observed in our previous paper[20], are omitted from this figure for clarity. **b** Crystal structure of $b$MF$_1$ with IF$_1$ bound to the αβ$_{DP}$ subunit (PDB: 2v7q). α$_{DP}$, β$_{DP}$, γ, and IF$_1$ are colored by dark red, pink, blue, and green, respectively. The β$_{DP}$ subunit has been omitted for clarity in the enlarged figure. **c** An illustration of the single-molecule rotation assay system of $b$MF$_1$. The stator α$_3$β$_3$-ring is immobilized on a glass surface. A magnetic bead (φ - 300 nm) is attached to the rotor γ subunit as a rotation probe via biotin-streptavidin interaction. Magnetic tweezers, consisting of two sets of electromagnets, were equipped onto the sample stage of the microscope.

of F$_1$[25–30]. The most common mechanism universally found in bacterial and mammalian F$_1$ is ADP inhibition in which F$_1$ spontaneously lapses into a resting state during rotation[31–33]. Mitochondrial F$_1$s have a unique inhibitor protein, termed the ATPase inhibitory factor 1 (IF$_1$)[34]. It is found in eukaryotic cells and is highly conserved particularly among mammalian cells[25]. Unlike the ε subunit[9,35–38] that some of the bacterial F$_1$s employ for inhibition, IF$_1$ is not a built-in inhibitor, but a dissociative one that associates with F$_1$ when pH decreases to acidic state or F$_1$ is isolated before association with F$_o$ to form the whole complex of F$_o$F$_1$[6]. The most remarkable feature of IF$_1$ is its condition-dependent manner of inhibition: IF$_1$ inhibits ATP hydrolysis activity of F$_1$ almost completely under ATP hydrolysis conditions, whereas, in ATP synthesis conditions, IF$_1$ dissociates from F$_1$ and does not interfere with ATP synthesis reactions after dissociation[34,39]. Notably, a few recent studies claim that IF$_1$ also suppresses the rate of ATP synthesis when IF$_1$ is overexpressed in cells[40,41].

The atomic details of the interaction between IF$_1$ and $b$MF$_1$ have been investigated in structural studies[42–44]. Full-length bovine mitochondrial IF$_1$ forms a homodimer complex via an antiparallel coiled coil[45], which associates two molecules of the F$_1$-ATPase with each N-terminal region, whereas the C-terminal region of IF$_1$ forms a coiled-coil structure for dimerization[42]. Deletion of the C-terminal residues 61–84 produces a stable monomeric form of IF$_1$, which achieves full inhibition capacity irrespective of pH change[46]. This simple platform of IF$_1$ is often used in biochemical[47,48] and structural studies[43,44], including this study. The crystal structure of the $b$MF$_1$-IF$_1$ complex[44] showed that the long α-helical structure in the middle section of IF$_1$ was bound to the interface of the αβ$_{DP}$ pair, whereas the short helix of the N-terminus was in contact with the γ subunit. The N-terminal short helix was linked to the middle-long helix via a distinct kink (Fig. 1b). The crystal structure of $b$MF$_1$ with three IF$_1$ units, referred to as $b$MF$_1$-(IF$_1$)$_3$, has been resolved[43], where each αβ interface was bound with IF$_1$. Because the structures of the αβ interface are different, IF$_1$'s show different conformational states. The structure of IF$_1$ on αβ$_{DP}$ is consistent with that of the 1:1 $b$MF$_1$-IF$_1$ complex, whereas the N-terminal short helix of IF$_1$ is not resolved on αβ$_{TP}$. The structure of IF$_1$ on αβ$_E$ shows the second half of a long helix,

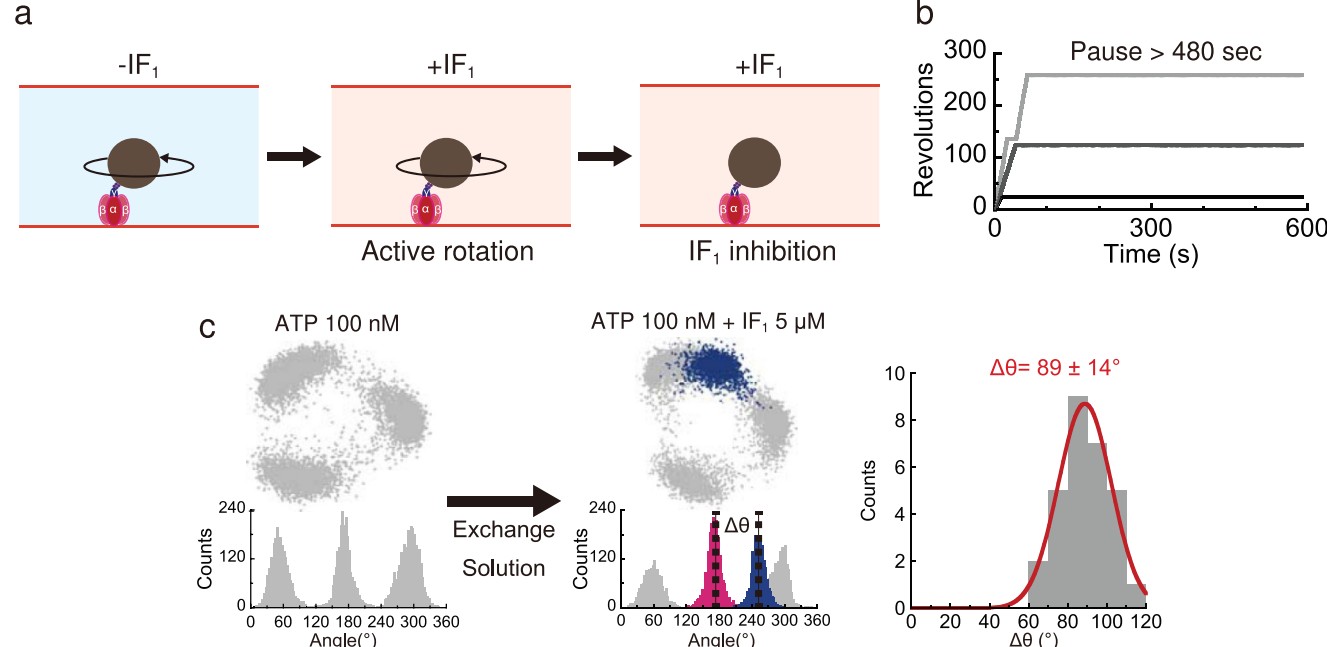

**Fig. 2 | Single-molecule analysis of IF$_1$-inhibited $b$MF$_1$. a** Experimental method. After observing a rotating $b$MF$_1$ molecule in IF$_1$-free solution, we introduced a mixed solution with IF$_1$ into the reaction chamber. The $b$MF$_1$ continued to rotate for a while but finally stopped rotation. **b** Typical time courses of $b$MF$_1$ in the presence of 3 μM IF$_1$ and 1 mM ATP. See Supplementary Fig. 2 for more detailed analyses. Source data are provided in the Source Data file. **c** Stall positions of IF$_1$ inhibition. (*Left*) An example of the IF$_1$-inhibited pauses. After observing the ATP-binding waiting dwell at 100 nM ATP, 5 μM IF$_1$ with 100 nM ATP was introduced into the reaction chamber. Blue data points represent stalls of IF$_1$ inhibition. (*Right*) The angular distance (Δθ) of IF$_1$-inhibited states from the left-side ATP-binding waiting dwell (pink) ($N$ = 29 pauses). Values represent mean ± SD estimated from Gaussian fitting of the plots. Source data are provided in the Source Data file.

and the remaining was unsolved. Based on these observations, the progressive folding of IF$_1$ coupled with γ rotation has been proposed. Progressive folding process has also been suggested in biochemical studies[48,49], although the progressive processes are simplified as a two-step process: the initial binding process and the subsequent isomerization process.

Compared to studies under hydrolysis conditions[48,50,51], studies on IF$_1$ under synthetic conditions are less advanced, despite its importance for the understanding of the condition-dependent inhibition mechanism of IF$_1$. IF$_1$-inhibited state of F$_1$-ATPase is known to be so stable that F$_1$ is unable to eject IF$_1$ by thermal agitation alone[48]. A typical condition for the unlocking from IF$_1$ inhibition is to charge sufficiently high *pmf* to membrane vesicles containing IF$_1$-inhibited F$_O$F$_1$[52–57]. However, the principal mechanism for IF$_1$ ejection from the catalytic site of F$_1$, followed by the recovery of catalysis in F$_1$ (hereafter, "activation from IF$_1$ inhibition" in this paper), remains elusive. Many fundamental questions are unsolved such as 'Does reversible rotation lead to activation from IF$_1$ inhibition?', 'Are there factors to enhance the activation?', and 'Which interactions among IF$_1$ and F$_1$ are responsible for IF$_1$ inhibition and for condition-dependent inhibition?'.

This study investigates the experimental conditions required for the dissociation of IF$_1$ from the inhibition complex using magnetic tweezers, which enable control of the angular orientation of the rotor during single-molecule rotation assays for $b$MF$_1$ (Fig. 1c). Similar to the mechanical activation of F$_1$ from the ADP-inhibited form, we forcibly rotate the rotor of IF$_1$-inhibited F$_1$, and define activation as the resumption of F$_1$ molecule rotation. The activation probability is determined as a function of the angle, similar to our previous stall-and-release experiments[20,21,33,58]. Further, we investigate the roles of the N-terminal short helix and the middle-long helix in inhibition and activation. These results highlight the molecular mechanism of IF$_1$ dissociation, which is critical for the unidirectional inhibition mechanism of IF$_1$.

## Results

### IF$_1$-inhibited states of $b$MF$_1$

The rotation of $b$MF$_1$ was monitored using magnetic beads (beads diameter, φ ~300 nm) attached to the γ subunit as a rotation probe and recorded at 30 frames per second (fps). During rotation, $b$MF$_1$ showed a transient pause in the absence of IF$_1$[20]. This was attributed to ADP inhibition. Supplementary Fig. 1 summarizes the kinetic analyses of ADP inhibition. Although it is dependent on ATP concentrations, the mean time for ADP inhibition range between 10−30 s. For IF$_1$ inhibition, monomeric bovine IF$_1$ (Δ61−84) was used, as described in previous studies[47,48].

The experimental procedure for the analysis and manipulation of IF$_1$-inhibited $b$MF$_1$ molecules was as follows (Fig. 2a); after identification of rotating particles in the presence of 1 mM Mg-ATP, a solution containing IF$_1$ and Mg-ATP was gently introduced into the flow cell. After rotations for several tens of seconds, all $b$MF$_1$ molecules stopped rotation without exception (Fig. 2b), and none of the observed molecules resumed rotation during the observation time of 480 s (Supplementary Fig. 2a). The pause duration was evidently longer than the duration of ADP inhibition, and thus attributed to IF$_1$ inhibition. Under 3 μM IF$_1$ and 1 mM ATP, the mean time to lapse into IF$_1$ inhibition was approximately 20 s (Supplementary Fig. 2b). The time constant for IF$_1$ inhibition obtained in our biochemical experiments at saturated ATP and IF$_1$ concentrations (Supplementary Fig. 9, gray) was 30 s, which is approximately 1.5 times longer than that in the single-molecule rotation assay. This is probably due to differences in experimental conditions; the single-molecule rotation assay selectively analyzes actively rotating molecules, whereas biochemical measurement is based on ensemble averaging of active and ADP-inhibited molecules. The latter would require a longer time to lapse into IF$_1$ inhibition than actively rotating molecules when ADP-inhibited form is off the pathway to IF$_1$ inhibition.

Next, we investigated the dwell position of $b$MF$_1$ when inhibited by IF$_1$. Experiments were performed under 100 nM ATP condition,

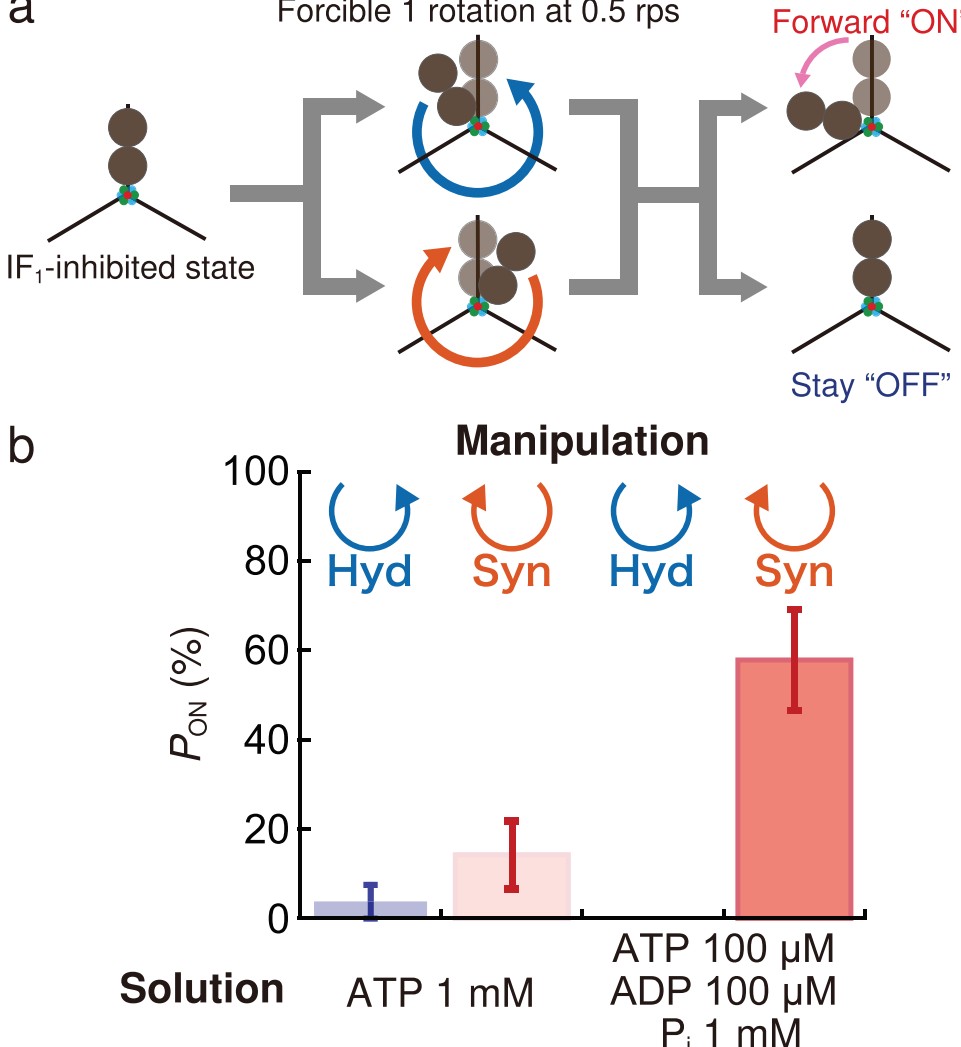

**Fig. 3 | Single-molecule manipulation of IF₁-inhibited $b$MF₁. a** Schematic images of the manipulation procedure. When $b$MF₁ was stalled by IF₁, the magnetic tweezers were turned on to stall $b$MF₁ to rotate one clockwise or counterclockwise revolution at the rate of 0.5 revolutions per second (rps). After manipulation, released $b$MF₁ either resumed its rotation (ON) or did not (OFF). These behaviors indicate whether or not IF₁ dissociates from $b$MF₁ under the stalling time.
**b** Reactivation probability of IF₁-inhibited $b$MF₁ after manipulation. "Hyd" and "Syn" represent the direction of hydrolysis (counterclockwise) and synthesis (clockwise), respectively. Values represent reactivation probability ($P_{ON}$) ± SD. $P_{ON}$ was defined as the probability of an ON event against total molecules. The SD of $P_{ON}$ is given as $\sqrt{P_{ON}(100 - P_{ON})/N}$, where $N$ is the number of total molecules ($N$ = 19-26 molecules). Source data and the exact number of molecules in each data point are provided as the Source Data file.

where $b$MF₁ showed three distinct pauses at the ATP-binding dwell angles (Fig. 2c). Transient pauses, such as ADP inhibition, are rarely observed[20]. When a solution containing 100 nM ATP and 5 μM IF₁ was introduced into the flow chamber, $b$MF₁ stopped rotating completely. The time constant for IF₁ inhibition at 100 nM ATP was 461 s (Supplementary Fig. 3), which was longer than that obtained with 1 mM ATP, i.e., 19.6 s (Supplementary Fig. 2b). The observed ATP-dependent effect of IF₁ inhibition was consistent with previous biochemical analysis[48,49,59,60]. Fig 2c shows a representative dataset for the angular position analysis of IF₁ inhibition. The angular position of IF₁ inhibition (Fig. 2c, blue) was determined using the ATP-binding dwell angles as the reference; it was defined as the angular distance from the left-side ATP-binding pause (Fig. 2c, pink). The position of IF₁ inhibition was determined as 89 ± 14° from the ATP-binding angle. This value was almost identical to the pause positions of $b$MF₁ stalled by AMP-PNP (76°)[20] and sodium azide (79° in Supplementary Fig. 4), which corresponds to the position of the catalytic dwell (80°). Thus, we confirmed that IF₁-inhibited $b$MF₁ pauses at the catalytic dwell angle, as seen in human mitochondrial F₁[19].

## Activation of IF₁-inhibited $b$MF₁ via forcible rotation

In the rotation assays, IF₁-inhibited $b$MF₁ did not resume the rotations once it lapsed into IF₁ inhibition. By contrast, it was reported that F₀F₁ is activated from IF₁ inhibition when the *pmf* is charged on the vesicle membrane, in which the F₀F₁ is embedded[55]. To explore the crucial conditions and factors for unlocking from IF₁ inhibition, IF₁-inhibited $b$MF₁ molecules were forcibly rotated using magnetic tweezers. First, we tested the clockwise rotation for one turn (Fig. 3a), considering that forcible rotation in the ATP synthesis direction for one turn is sufficient for the unlock from IF₁ inhibition. Before the forcible rotation, the buffer in a flow cell was exchanged with IF₁-free ATP solution (1 mM) to prevent possible rebinding of IF₁. However, the molecules did not show activation in most cases (Fig. 3b). Only a small fraction of molecules (10%) resumed continuous rotation. When forcibly rotated in the counterclockwise direction, the activation probability was even lower (<4%), suggesting the rotational-direction-dependence for the activation from the IF₁ inhibition. The reactivation is a unique phenomenon that was never observed without

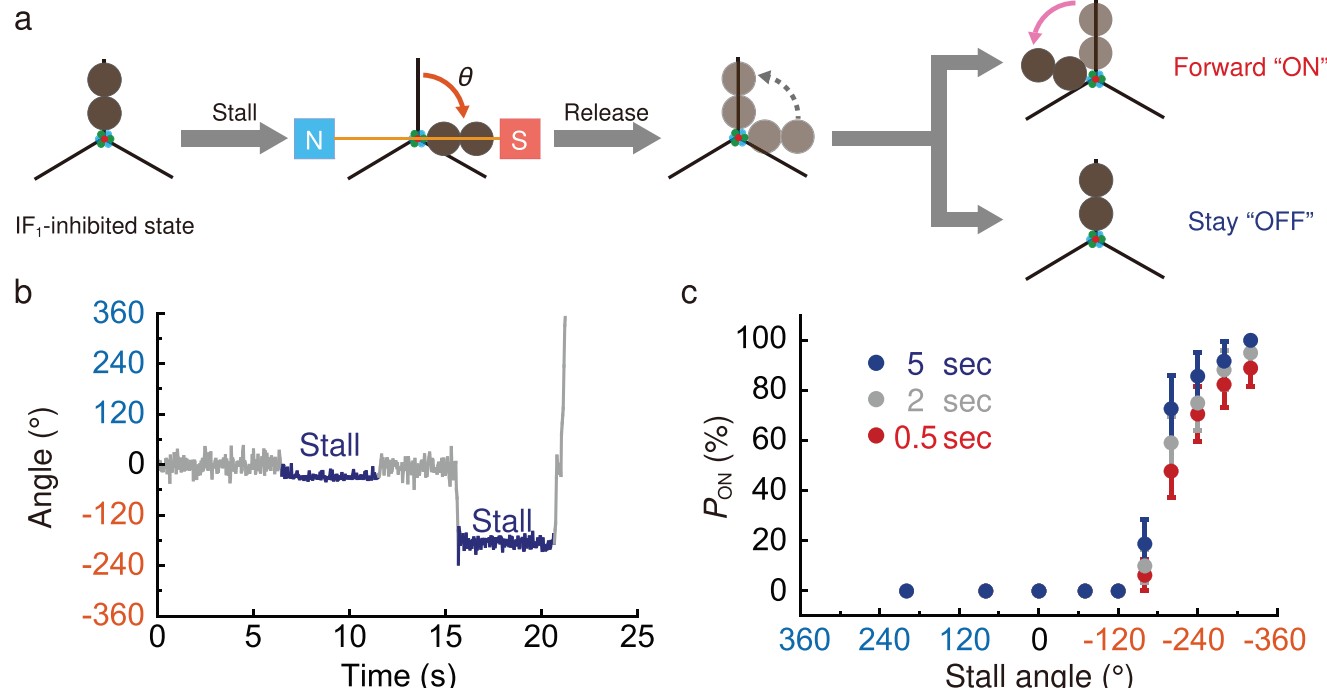

**Fig. 4 | The stall-and-release experiment of IF$_1$-stalled $b$MF$_1$. a** Schematic images of the manipulation procedure in the stall-and-release experiment. When $b$MF$_1$ was stalled by IF$_1$, the magnetic tweezers were turned on to stall $b$MF$_1$ at the target angle. After the set time had elapsed, the magnetic tweezers were turned off, and the molecule returned to the initial angle. Released $b$MF$_1$ either resumes its rotation (ON) or stays at the initial position (OFF). These behaviors indicate whether or not IF$_1$ dissociates from $b$MF$_1$ under the stalling time. **b** A representative time course of the stall-and-release experiment under 100 μM ATP, 100 μM ADP, and 1 mM P$_i$ in the presence of 3 μM IF$_1$. In this figure, the stall time for both trials (blue) was 5 s and the stall angles were -26° and -168°, respectively. **c** Angle dependence of

reactivation probability under 100 μM ATP, 100 μM ADP, and 1 mM P$_i$ in the presence of 3 μM IF$_1$. Each data point was obtained from 10 to 57 trials using 4 to 14 molecules. Counterclockwise rotation (blue) and clockwise rotation (orange) are defined as positive and negative direction, respectively. The colors on the plots represent the stall time of 0.5 s (red), 2 s (gray) and 5 s (blue), respectively. Values represent reactivation probability ($P_{ON}$) ± SD. $P_{ON}$ was defined as the probability of an ON event against total trials. The SD of $P_{ON}$ is given as $\sqrt{P_{ON}(100 - P_{ON})/N}$, where $N$ is the number of total trials in each data point. Source data and the exact number of trials in each data point are provided in the Source Data file.

manipulation with magnetic tweezers, but the probability is too low compared to the reported *pmf*-induced full activation of F$_o$F$_1$[52–57].

Next, we tested the effect of ADP and P$_i$ in a forcible rotation to further mimic ATP synthesis conditions. After confirming IF$_1$ inhibition, the buffer in the flow cell was gently exchanged with IF$_1$-free ATP synthesis buffer (100 μM ATP, 100 μM ADP, 1 mM P$_i$), and then, IF$_1$-inhibited $b$MF$_1$ molecules were forcibly rotated in the clockwise or counterclockwise direction for 360°. As shown in Fig. 3b, a remarkable increase in activation was observed when IF$_1$-inhibited $b$MF$_1$ was rotated in the clockwise direction in the presence of ADP and P$_i$; the probability of activation was about 60%. This is in contrast to the reactivation probability for the clockwise manipulation in ADP/P$_i$-free solution (10%). An evident rotational-direction-dependent manner was observed; the counterclockwise rotation failed the activation. These observations indicate that both directional manipulation and the presence of substrates for ATP synthesis are requisite for the efficient activation of the IF$_1$-inhibited state. To investigate the effect of individual substrates on activation, we performed the same manipulation experiment in the presence of ADP or P$_i$, respectively (Supplementary Fig. 5). P$_i$ was more effective for reactivation (28%), whereas the activation achieved only 4% in the presence of ADP. Notably, once activated, $b$MF$_1$ molecules did not show long pauses that are attributable to IF$_1$ inhibition in IF$_1$-free solution. This observation indicates that activation is accompanied by dissociation of IF$_1$ from F$_1$.

**Angle-dependence of IF$_1$ dissociation**

We determined the activation probability as a function of the rotation angle by performing a "stall-and-release" experiment. The experimental procedure was as follows (Fig. 4a); after confirming IF$_1$

inhibition, the inhibited $b$MF$_1$ molecules were rotated to stall at the target angle for the programmed period of 0.5–5.0 s with the magnetic tweezers. After the set time had elapsed, the magnetic tweezers were turned off to release the F$_1$ molecule. The released molecule showed two types of behaviors (Fig. 4b): starting active rotation or returning to the initial position of IF$_1$ inhibition to resume the pause. The former behavior was identified as activation from IF$_1$ inhibition, and the latter as failure of activation. In contrast to the abovementioned activation by a forcible 360° rotation, we conducted a stall-and-release experiment without washing IF$_1$ in solution. The molecules again lapsed into IF$_1$ inhibition after activation due to the rebinding of IF$_1$ in a solution containing 3 μM IF$_1$. The mean rotation time until the re-inactivation of activated F$_1$ molecules was 18.3 s (Supplementary Fig. 6), and the result is consistent with the time constant of IF$_1$ inhibition (15.2 s), indicating that re-inactivation was due to the rebinding of IF$_1$ from solution. We repeated the manipulation of each molecule to confirm reproducibility. After manipulation, a few molecules occasionally exhibited unusual behaviors, such as random tethered Brownian motion, and nonspecific binding to or detachment from the coverslip. These data were omitted from the analyses. A detailed description of the analysis of molecules in this experiment is presented in Supplementary Note and Supplementary Fig. 7.

Experiments were conducted with 100 μM ATP, 100 μM ADP, and 1 mM P$_i$. Fig 4c shows the experimental results, where the counterclockwise direction of the stall angle was defined as positive relative to the initial IF$_1$-inhibited state. In principle, activation was rarely observed in the counterclockwise manipulation, which is consistent with the experimental results for forcible 360° rotation manipulation (Fig. 3b). For the clockwise manipulation, rotation up to 120° did not

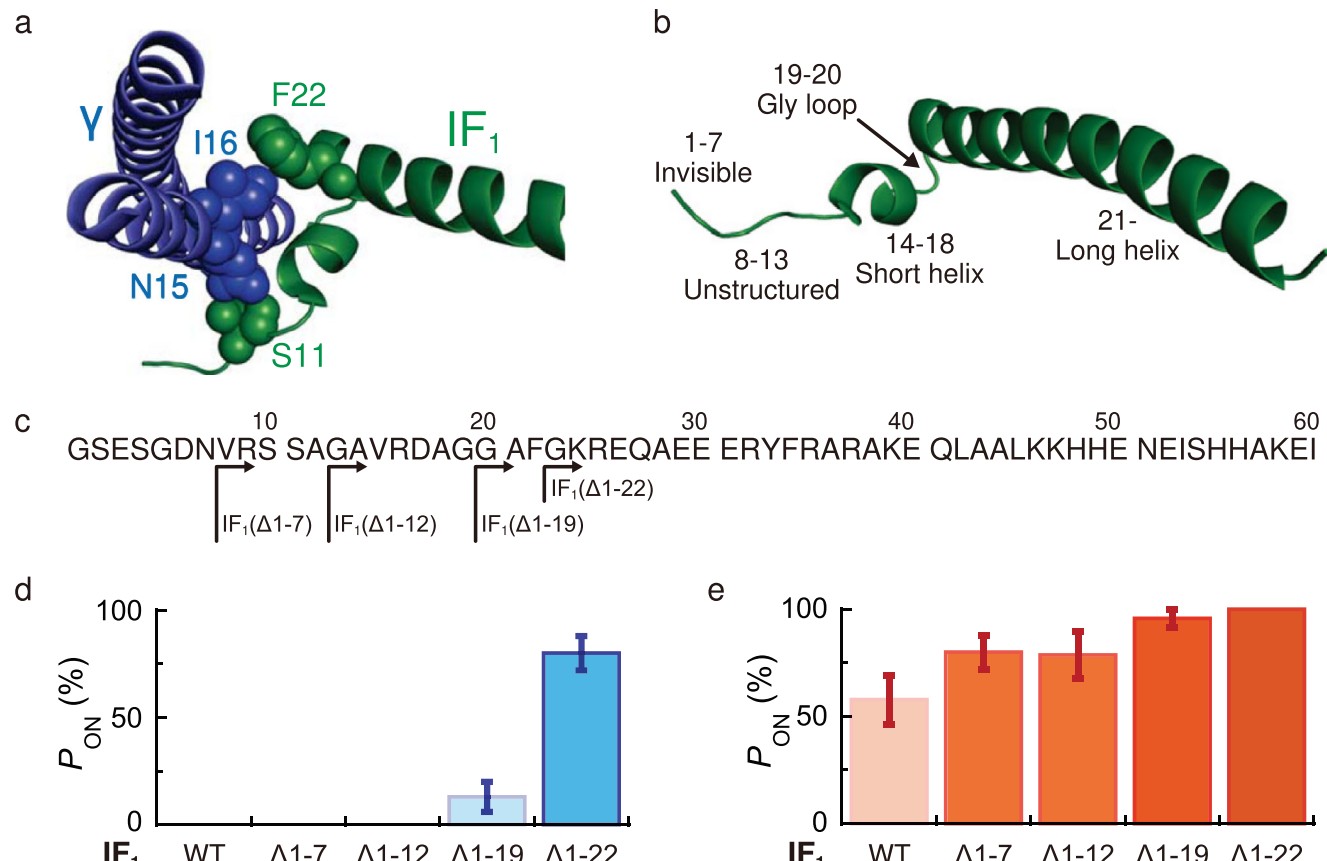

**Fig. 5 | Analysis of mutant IF₁'s with the N-terminal truncation. a** Enlarged illustration of the interactions between IF₁ and γ subunit in the $b$MF₁-IF₁ complex (PDB: 2v7q). S11 and F22 in IF₁ (green) can interact with N15 and I16 in γ subunit (blue), respectively. **b** Details of the IF₁ structure (PDB: 2v7q), where residues 8–50 are resolved. Residues 1–7 are not resolved. Residues 8-13 are resolved and form an extended structure. Residues 14–18 and 21–50 form short and long helix, respectively, linked by glycine loop in residues 19–20. **c** Sequence of bovine IF₁$^{1-60}$ and definition of N-terminal truncated mutants. **d, e** Reactivation probability of

inhibited $b$MF₁ by mutant IF₁s after **d** counterclockwise and **e** clockwise manipulation. Experiments were performed under 100 µM ATP, 100 µM ADP, and 1 mM Pᵢ. The data for IF₁(WT) are also plotted for comparison (Fig. 3b). Values represent reactivation probability ($P_{ON}$) ± SD. $P_{ON}$ was defined as the probability of an ON event against total molecules. The SD of $P_{ON}$ is given as $\sqrt{P_{ON}(100 - P_{ON})/N}$, where $N$ is the number of total molecules ($N$ = 11–25 molecules). Source data and the exact number of molecules in each data point are provided in the Source Data file.

activate IF₁-inhibited $b$MF₁. When rotated over 200°, the activation probability, i.e., $P_{ON}$ was remarkably increased. $P_{ON}$ was 60% when stalled at −200° for 2 s, and reached over 90% when stalled at −320° for 5 s. At each stall angle, $P_{ON}$ increased with stall times. These features are similar to those observed in our previous studies on the angle dependence of ATP binding and hydrolysis in TF₁[58] and $b$MF₁[20] as well as the angle dependence of activation from ADP inhibition[32]. These results on angle-dependent activation are discussed in detail in the *Discussion* section.

**N-terminal-truncated mutants of IF₁**

With an aim to investigate which structural elements of IF₁ are responsible for the observed rotation-direction-dependent activation, we generated IF₁ mutants with N-terminal truncations and tested their inhibitory functions in biochemical and single-molecule manipulation experiments (Fig. 5). IF₁ consists of the N-terminal region that is missing in the crystal structure (1–7), unstructured region (8–13), the short helix region (14–18), glycine loop (19–20), and the long helix region (21–60) (Fig. 5a, b). The long helix of IF₁ mainly interacts with the βDP subunit, whereas the proximity contacts with the γ subunit are found at S11 and F22 (Fig. 5a)[43]. We prepared four truncation mutants of IF₁: IF₁(Δ1–7), IF₁(Δ1–12), IF₁(Δ1–19), and IF₁(Δ1–22) (Fig. 5c). The inhibitory effects of the mutants were first analyzed using a solution ATPase assay (Supplementary Figs. 8 and 9). IF₁(Δ1–7) and IF₁(Δ1–12) showed similar inhibitory effects to IF₁(WT); upon association with

$b$MF₁, the mutants of IF₁ completely inhibited ATPase activity. In contrast, IF₁(Δ1–19) and IF₁(Δ1–22) showed a reduced inhibitory effect. At concentrations lower than 0.5 µM, IF₁(Δ1–19) did not completely inhibit ATPase activity, suggesting a reversible association/dissociation. IF₁(Δ1–22) showed a lower inhibitory effect with a peculiar kinetic behavior. Upon addition of IF₁(Δ1–22) into the assay mixture, ATPase activity decreased, followed by slow recovery. Although the mechanisms underlying such complex behavior are unclear, these biochemical measurements showed that the inhibitory effect of IF₁(Δ1–19) and IF₁(Δ1–22) was less than that of IF₁(Δ1–7) and IF₁(Δ1–12).

We analyzed the rotation of $b$MF₁ in the presence of truncated mutants of IF₁. Similar to the biochemical experiments, we observed that IF₁(Δ1–7) and IF₁(Δ1–12) showed nearly the same properties as those of IF₁(WT) (Supplementary Fig. 10). As shown in the time course and pause time analysis, inhibition by these mutants was essentially irreversible; IF₁-inhibited $b$MF₁ did not spontaneously resume active rotations within the observation time. In contrast, further N-terminal truncation had a significant effect on IF₁ inhibition. With IF₁(Δ1–19) and IF₁(Δ1–22), $b$MF₁ molecules did not completely stop the rotation, but the molecules showed frequent transitions between rotation and intervening pauses (Supplementary Fig. 10j and m). The mean duration of pauses with IF₁(Δ1–19) or IF₁(Δ1–22) was 250 s or 60 s, respectively. These observations show that the short helix and N-terminus of the long helix of IF₁ have critical roles in the stabilization of IF₁ inhibition.

Rotation-direction-dependent activation was investigated with the mutant $IF_1$'s. Whereas the activation experiments for $IF_1(\Delta 1-7)$ and $IF_1(\Delta 1-12)$ were conducted in $IF_1$-free solutions, those were done in the presence of $IF_1$ for $IF_1(\Delta 1-19)$ and $IF_1(\Delta 1-22)$. This is because these mutants can dissociate from $F_1$ during buffer exchange, hampering the activation experiment. Fig 5d and e show the activation probabilities after a forcible 360° rotation in the counterclockwise and clockwise directions, respectively. The probabilities of activation from inhibition with $IF_1(\Delta 1-7)$ and $IF_1(\Delta 1-12)$ showed similar trends to that of $IF_1(WT)$ in both directions; i.e., no activation after counterclockwise rotation, whereas clockwise rotation induced activation with a significant probability of 80%, which is higher than that for $IF_1(WT)$ at 58%. These results indicate that the unstructured N-terminal region from positions 1 to 13 of $IF_1$ is not responsible for the principal mechanism of $IF_1$ inhibition, although the region contributes to the structural stabilization of the $bMF_1$-$IF_1$ complex. Unlike the truncation of the unstructured N-terminal region, the truncation of the short helix and the subsequent N-terminal region of the long helix had the distinctive impact on rotation-direction-dependent activation. For the mutants $IF_1(\Delta 1-19)$ and $IF_1(\Delta 1-22)$, the activation probabilities after clockwise or counterclockwise rotation were both higher than those for $IF_1(WT)$, $IF_1(\Delta 1-7)$, and $IF_1(\Delta 1-12)$. In particular, a significant fraction of events showed activation after counterclockwise rotation: 13% for $IF_1(\Delta 1-19)$ and 80% for $IF_1(\Delta 1-22)$. The activation probability after clockwise rotation was also higher than that for $IF_1(WT)$, $IF_1(\Delta 1-7)$, and $IF_1(\Delta 1-12)$, and was close to 100%. These results suggest that $IF_1$ readily dissociates from $bMF_1$ without the short helix and the N-terminal tip of the long helix, as suggested by biochemical data. Importantly, $IF_1(\Delta 1-22)$ almost loses the rotation-direction-dependent feature, and the activation probabilities after clockwise or counterclockwise rotation are more than 80%. Considering that $IF_1$ shows direct contact with the γ subunit in the truncated regions, these results suggest that the rotation-direction-dependent dissociation mechanism is based on contact with the γ subunit. One may consider that $IF_1(\Delta 1-22)$ dissociates from $bMF_1$ not by forcible rotation, but by its nature of spontaneous dissociation. However, the manipulation time is 2 s which is too short for spontaneous dissociation of $IF_1(\Delta 1-22)$; the probability of spontaneous dissociation within 2 s is only 3% when estimated from the mean time constant of spontaneous activation, 60 s (see Supplementary Note). Thus, the observed activation with $IF_1(\Delta 1-22)$ results due to forcible rotation.

## Discussion

In this study, we observed the activation from $IF_1$ inhibition by forcible rotation with magnetic tweezers. Activation is accompanied by the ejection of bound $IF_1$ from $bMF_1$. This is supported by the following observations: re-inactivation by $IF_1$ was observed only when the solution contained $IF_1$. In the absence of $IF_1$, the activated molecules did not show long pauses attributable to $IF_1$ inhibition. Second, the mean rotation time before re-inactivation was in good agreement with that for $IF_1$ inhibition (Supplementary Fig. 6). The analysis of activation from $IF_1$ inhibition provides important implications for the mechanism of $IF_1$ inhibition. $bMF_1$ molecules during $IF_1$ inhibition were activated when the γ subunit was forcibly rotated with magnetic tweezers for over 200° in the clockwise (synthesis) direction. The activation probability was significantly enhanced in the presence of $P_i$ in the solution by a factor of 15, although ADP also had an impact on the activation by a factor of 2 or more (Supplementary Fig. 5). Thus, it is evident that reactivation from $IF_1$ inhibition occurs under ATP synthesis conditions. With $IF_1(WT)$, the probability of activation reached 100% when stalled at −320° for 5 s. This is sufficiently high to explain the *pmf*-induced activation of the $IF_1$-inhibited $F_oF_1$[52–57]. Thus, the principal mechanism for the *pmf*-induced activation of $IF_1$-inhibited $F_oF_1$ is the ejection of $IF_1$ from $F_1$ by forcible rotation powered by *pmf*-driven $F_o$ motor in the $F_oF_1$ complex. The requirement of ADP and $P_i$ for efficient activation

indicates that forcible clockwise rotation should be coupled with the ATP synthesis reaction for $IF_1$ ejection. As the presence of substrates enhances the cooperative nature of $F_1$, $IF_1$ may likely be ejected through a concerted conformational transition of the α and β subunits coupled with γ subunit rotation.

We observed a significant increase in the probability of activation when the $IF_1$-inhibited $bMF_1$ was rotated over −200° (Fig. 4c). The crystal structure of $bMF_1$-$IF_1$ complex show that $IF_1$ binds to $\beta_{DP}$ which represents the +200° state from the ATP-binding state (0°) in the scheme. These results suggest that $IF_1$ is ejected through the state transitions of the β subunit from the +200° state to the 0° state or −40° (Fig. 6a). This state transition should be coupled with the conformational transition of the β subunit from a closed to an open conformation. The closed-to-open transition accompanies the swing-out motion of the C-terminal domain of the β subunit to which $IF_1$ binds via the long α helix. Thus, $IF_1$ dissociates from the β subunit in an open conformation that facilitates $IF_1$ dissociation. The closed-to-open conformational transition would destabilize the $bMF_1$-$IF_1$ complex by pulling the N-terminal regions of $IF_1$ out of the γ subunit, because the C-terminal domain of the β subunit in the open conformation is apart from the axis of the γ subunit, compared with the closed conformation. This dissociation model is almost the reverse process of the $IF_1$ association proposed for the $bMF_1$-$(IF_1)_3$ structure, which suggests the progressive folding of $IF_1$ coupled with the conformational transition of the β subunit from $\beta_E$ to $\beta_{DP}$ via $\beta_{TP}$.

Estimating the energy required for activation from $IF_1$ inhibition is important to discuss under what conditions ATP synthase is activated in the cell. Although this study does not measure the torque of magnetic tweezers applied to $F_1$ molecules, rough estimation is possible based on several assumptions. When $bMF_1$ generates torque of 40 pN·nm like other $F_1$s, the minimum value for proton motive force as voltage to reverse $F_1$ is 196 mV, considering the proton stoichiometry of 8 protons/turn. In addition, the minimum energy for $IF_1$ ejection is assumed to be 140 pN·nm (84 kJ/mol).

The above discussion indicates the reversible association/dissociation processes of $IF_1$. This can be viewed as an inevitable property for $IF_1$ to avoid being Maxwell's demon that violates the second law of thermodynamics by allowing only one direction of motion in a microscopic system. If $IF_1$ can dissociate from $F_1$ without coupling with clockwise rotation, while $IF_1$ association is tightly coupled with counterclockwise rotation, the second law of thermodynamics will be violated as Maxwell's demon. This is evident when one considers the rotary motion of $F_oF_1$ under conditions where the torques of $F_1$ and $F_o$ are balanced. In $IF_1$-free conditions, the rotor complex should show non-biased Brownian motion in both directions: clockwise and counterclockwise, in the $F_oF_1$ complex. Even though the rotor can cause occasional rotary steps in both directions, the mean rotational displacement does not increase. In contrast, in the presence of $IF_1$, $F_1$ should make a +200° rotation coupled upon the association of $IF_1$. This is because 'the tight coupling of $IF_1$ association and the rotation' means that $IF_1$ association induces +200° rotation. Although the pause by $IF_1$ inhibition should be long, $IF_1$ should eventually dissociates from $F_1$. Along the above assumption, $IF_1$ is dissociated without biasing rotation. Thus, each cycle of association and dissociation of $IF_1$ should cause the rotation by +200° in hydrolysis direction. After multiple events of $IF_1$ association and dissociation, $F_1$ undergoes unidirectionally biased rotation in hydrolysis direction. Thus, the assumption of the asymmetric coupling model should violate the second law of thermodynamics, and the reversibility of $IF_1$ association/dissociation seems inevitable. For the same reason, the requirement for ADP and $P_i$ for $IF_1$ dissociation is also inevitable, considering that $IF_1$ association is coupled with ATP hydrolysis.

This consideration would be applicable to other regulatory systems of $F_oF_1$. In addition to $IF_1$, several types of inhibitory mechanisms can block the ATP hydrolysis. The ζ subunit of α-proteobacteria

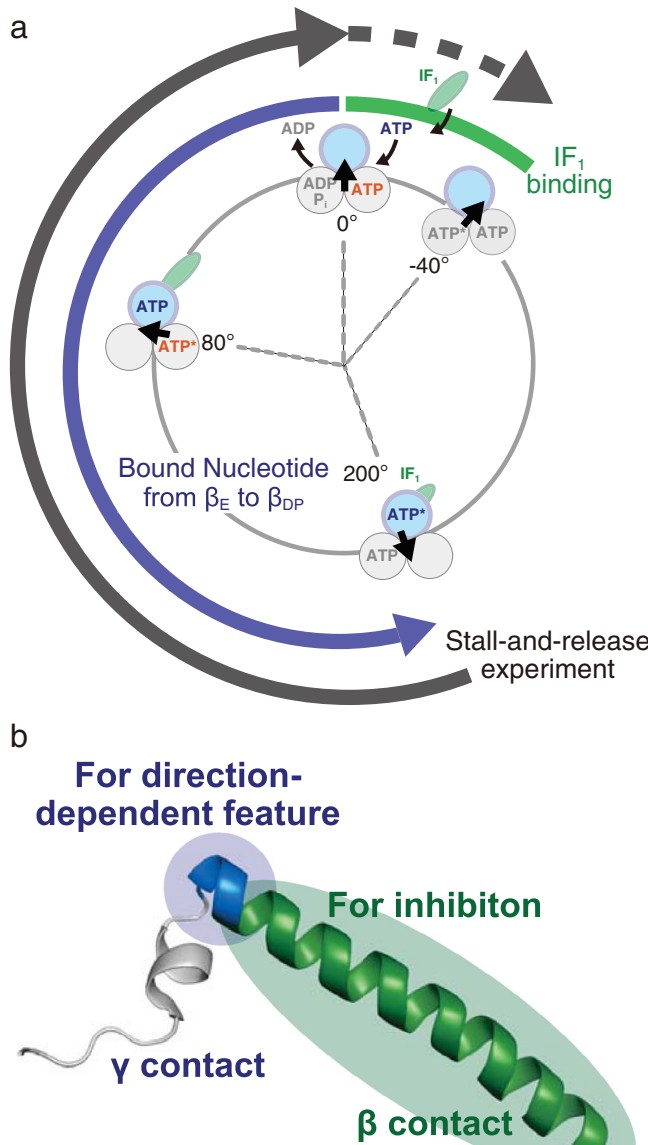

a

b

**For direction-dependent feature**

**For inhibiton**

γ contact

β contact

**Fig. 6 | The proposed mechanisms presented in this study. a** Coupling scheme of IF$_1$ dissociation with the rotary mechanism in $b$MF$_1$. A part of the reaction scheme of rotary catalysis in $b$MF$_1$ (Fig. 1a) is described with only the essential elements. The green oval represents IF$_1$. The green line in the scheme represents the IF$_1$ binding angle in the range of −40°-0°. The blue line represents the angular difference (0°-200°) required for the conformational transition of the blue β subunit from β$_E$ to β$_{DP}$ via β$_{TP}$. The black line represents the angle distance for IF$_1$ dissociation, estimated in the stall-and-release experiment. **b** Schematic representation of the role of IF$_1$ (PDB: 2v7q). The N-terminal region of IF$_1$ wraps around the γ subunit, and the long helix of IF$_1$ interacts with the β$_{DP}$ subunit. In addition to the N-terminal amino acid residues, including the short helix (residues 14-18), residues 20–22 (blue) play a critical role in the rotation-direction-dependent activation. Residues starting at residue 23 (green) in the central long helix work as a prototypical inhibitor.

inhibits hydrolysis by binding to the interface of the αβ$_{DP}$ pair in a manner similar to IF$_1$[61–63]. Mycobacteria have a specific insertion sequence at the C-terminal of the α subunit called the α-extension loop[64,65]. Recent cryo-EM analysis showed that the α-extension loop binds to a specific loop on the γ subunit, which is considered to block rotation in a counterclockwise direction[26,66]. When these inhibitory processes are coupled with the rotation of the γ subunit, the dissociation of the ζ subunit or the α-extension loop has to be coupled

with rotation in the opposite direction, according to the above contention.

All the experiments described in the main text were performed at room temperature; 23 ± 2 °C for single-molecule experiments and 25 °C for solution experiments. In order to confirm that IF$_1$ kinetics at the physiological temperature of bovine mitochondria (i.e., 37 °C) is essentially the same as that at room temperature, the biochemical assay of IF$_1$(WT), IF$_1$(Δ1-19) and IF$_1$(Δ1-22) at 37 °C was performed (Supplementary Figs. 13 and 14). As a result, the trends for the residual ATPase activity at 37 °C were principally identical to that at 25 °C as described in Supplementary Fig. 8: IF$_1$(WT) showed complete inhibition with almost no residual activity, whereas IF$_1$(Δ1-19) and IF$_1$(Δ1-22) did not completely inhibit ATPase activity. These results suggest that the principle of IF$_1$ inhibition and dissociation does not differ between room temperature to physiological temperature.

Structural analyses of the $b$MF$_1$-IF$_1$ complex showed that IF$_1$ has interactions with $b$MF$_1$ via two parts: an N-terminal region with an unstructured loop and a short helix (1–20), and a central long helix (21–60) (Fig. 5a, b). The major interaction between IF$_1$ and F$_1$ is formed by the central long helix, which is accommodated on the C-terminal domain of the β subunit. The N-terminal region of IF$_1$ wraps around the γ subunit, with evident contacts at S11 and F22 of IF$_1$. Accordingly, two scenarios for the principal mechanism of IF$_1$ inhibition are possible[42]. First is the mechanical hindrance of the γ subunit rotation by the N-terminal region of IF$_1$. The second model assumes the prevention of β conformational transitions by the central long helix of IF$_1$. The investigation of IF$_1$ mutants with N-terminal truncation provides important clues to this question. We observed that IF$_1$(Δ1–19) and IF$_1$(Δ1–22) maintained inhibitory potency to halt the catalysis and rotation of $b$MF$_1$, although these mutants were remarkably less effective than IF$_1$(WT) or other mutants. IF$_1$(Δ1–22) was truncated before F22, losing all residues that were in contact or close proximity to the γ subunit. Thus, our study suggests that the interaction with the γ subunit is dispensable, supporting the latter model. Our result was also supported by the research using F$_1$ and IF$_1$ from yeast mitochondria: deletion of all the residues preceding F17 in yeast IF$_1$, corresponding to F22 in bovine IF$_1$, still maintained the inhibitory capacity[67]. This result implies a universal IF$_1$ inhibition mechanism beyond the species level.

In the atomic structures of $b$MF$_1$-IF$_1$ and $b$MF$_1$-(IF$_1$)$_3$, hydrophobic residues in the central long helix, including Y33, form strong interactions with the C-terminal domain of the β subunit, providing most of the binding energy. The binding of IF$_1$ to F$_1$ is further augmented by salt-bridge formation between E30 in IF$_1$ and R408 in the β$_{DP}$ subunit. The contribution of these residues was experimentally confirmed in mutagenetic approaches[47,48]. A possible molecular mechanism for IF$_1$ inhibition is that the tight binding of IF$_1$ with the β subunit prevents conformational transition required for rotary catalysis. There are several highly conserved charged residues in the C-terminus of IF$_1$, and the roles of these residues have not been clarified. As a preliminary trial, we conducted molecular dynamics simulation[68] where the γ subunit is forcibly rotated in synthesis direction. The interaction between the charged residues of the γ subunit and negative charge of IF$_1$ was suggested, which is reminiscent of the ionic track[69]. Based on this observation, we tested the role of these conserved charged residues by selectively substituting five residues with alanine (Supplementary Figs. 11 and 12). Except for a slight decrease in binding affinity to $b$MF$_1$, no clear differences were found between IF$_1$(WT) and the alanine-substitution mutant in the IF$_1$ inhibition assay as well as in manipulation assay. Our experiments suggest that C-terminal charged residues have little impact on the inhibitory mechanism and the rotation-direction-dependent dissociation.

Analyses of N-terminal truncation mutants revealed the molecular mechanism of rotary-direction-dependent dissociation of IF$_1$ from F$_1$ (Fig. 6b). IF$_1$(Δ1–22), which loses residues in contact with the γ

subunit, showed a significantly high activation probability after the forcible rotation of 360°. The important finding in this mutant is that both of the clockwise and counterclockwise rotation resulted in the efficient activation from $IF_1$ inhibition with almost equal probability (Fig. 5d, e). This is in contrast to the asymmetric features of $IF_1$(WT), $IF_1$(Δ1–7), and $IF_1$(Δ1–12), which showed activation only via clockwise rotation. In the case of $IF_1$(Δ1–19), the asymmetric effect of forcible rotation was retained. However, a higher activation probability is observed. These observations indicate that the N-terminal region of the central long helix plays a crucial role in rotary-direction-dependent activation, and the N-terminal short helix also contributes to this result. The truncated residues in $IF_1$(Δ1–22), in addition to the 1–19 truncation, are G20, A21, and F22. Among them, F22 is bulky compared with the other residues. Further, in the crystal structures of the $bMF_1$-$IF_1$ complex, F22 is in direct contact with I16 of the γ subunit. Considering these points, F22 is likely one of the most critical residues for rotary-direction-dependent activation. Molecular dynamics simulations of the $bMF_1$-$IF_1$ complex would provide more detailed information on the molecular mechanism of rotation-direction-dependent activation.

## Methods

### Purification of $bMF_1$ and $IF_1$

$F_1$-ATPase from bovine mitochondria with two mutated cysteine residues on A99 and S191 at the γ subunit and nine histidine residues (His-tag) at the N-terminus of the β subunit (hereafter referred to as $bMF_1$) was purified as described previously[20]. $IF_1^{1-60}$ with the linker and mScarlet fused to the C-terminus (referred to as $IF_1$(WT)) was purified as described previously[48]. The N-terminal truncation mutants and the C-terminal alanine substitution mutant of $IF_1$ were prepared as follows. Fragments of the N-terminal truncated mutants, deleted in the region encoding the corresponding N-terminal region, were generated by PCR using sets of mutation primers. The sequence encoding the C-terminal alanine mutant was amplified by PCR using primers containing mutations. The sequences of the PCR primers are provided in the Source Data file. The resulting PCR products were subjected to gel electrophoresis and purification. The insert plasmid and the vector $IF_1$(WT) plasmid were digested with the same restriction enzymes. The products were ligated and introduced into *Escherichia coli* JM109. The sequence of the recombinant $IF_1$ plasmid was confirmed using the Fasmac sequencing service (Fasmac, Japan). The purified mutant proteins were confirmed by SDS-PAGE and MALDI-TOF/TOF mass spectrometry (Genomine. Inc., Korea or IDEA Consultants, Inc., Japan) (Supplementary Fig. 15 and Supplementary Table 1).

### Solution experiment

The ATPase activity of $bMF_1$ was monitored as the rate of NADH oxidation[48]. The basal buffer contained 50 mM HEPES-KOH (pH 7.5), 50 mM KCl, 2 mM $MgCl_2$, and an ATP-regenerating system (0.2 mg/mL pyruvate kinase and 2.5 mM phosphoenolpyruvate) supplemented with 0.2 mM NADH and 50 μg/mL lactate dehydrogenase, as described previously[48]. Experiments were performed at 25°C, unless otherwise indicated, using a V-660 (JASCO, Tokyo, Japan) UV/VIS spectrophotometer equipped with a peltier-type temperature controller (ETCS-761). ATP hydrolysis by $bMF_1$ was initiated by adding purified $bMF_1$ to the basal buffer containing 1 mM ATP. For the reaction to reach a steady state, we waited for 180 s before injecting $IF_1$ into the reaction mixture. After the $IF_1$ injection, the rate of ATPase activity changed. Inhibition by $IF_1$ was quantified by estimating the apparent rate constants for $IF_1$ ($k_{inhibition}^{app}$), which were determined by fitting the decay using the following equation:

$$y(t) - y_0 = V_\infty t + \frac{V_0 - V_\infty}{k_{inhibition}^{app}} \left\{ 1 - \exp(-k_{inhibition}^{app} t) \right\} \qquad (1)$$

where $y(t)$ and $y_0$ are the absorbance values at the time $t$ and 0 after $IF_1$ injection into the solution cuvette, respectively, and $V_0$ and $V_\infty$ are the initial and final reaction rates, respectively. Fitting was performed using the time course of NADH absorbance at 2 s after $IF_1$ addition. Because no exponential decay was observed in the time course of $IF_1$(Δ1–19) and $IF_1$(Δ1–22) inhibition (Supplementary Figs. 8g, 13c, e), curve fitting using Eq. (1) was not performed. Residual ATPase activity was measured by comparing the activity just before $IF_1$ addition and 350 s after $IF_1$ addition. In Supplementary Figs. 8 and 11, values were plotted over intermediate or higher $IF_1$ concentrations ($K_M^{IF_1}$) to see equilibrium at saturating conditions.

The sequential $IF_1$ inhibition was analyzed using the following reaction scheme:

$$F_1 + IF_1 \underset{k_{off}}{\overset{k_{on}}{\underset{\longleftarrow}{\longrightarrow}}} F_1 \cdot IF_1 \overset{k_{lock}}{\rightarrow} F_1 \cdot IF_1^{lock}$$

where $F_1 \cdot IF_1^{lock}$ represents the inactive state, and $F_1 \cdot IF_1$ is the intermediate state that is still catalytically active. $k_{on}$ and $k_{off}$ represent the rate constants of association and dissociation, respectively. $k_{lock}$ is the rate constant of isomerization to the locked state. According to this scheme, $k_{inhibition}^{app}$ can be expressed as follows:

$$k_{inhibition}^{app} = \frac{k_{lock}[IF_1]}{K_M^{IF_1} + [IF_1]} \qquad (2)$$

$$K_M^{IF_1} \equiv \frac{k_{off} + k_{lock}}{k_{on}} \qquad (3)$$

which shows a typical hyperbolic relationship.

### Rotation assay of $bMF_1$

For the rotation assay, $bMF_1$ was immobilized on the glass surface, and magnetic beads were attached to the two cysteine residues of the γ subunit (γA99C and γS191C) through a biotin-avidin interaction. The protocol was as follows[20]. A flow chamber (~5 μL) was prepared with two coverslips (18 × 18 mm² at the top and 24 × 32 mm² at the bottom; Matsunami Glass) with double-sided tape (7602 #25, Teraoka) as a spacer. Purified $bMF_1$ (~500 pM) in observation buffer (50 mM HEPES-KOH (pH 7.5), 50 mM KCl, and 2 mM $MgCl_2$) was gently introduced into the flow chamber and incubated for 5–10 min. After washing unbound $bMF_1$ with more than 50 μL of observation buffer containing ~10 mg/mL BSA, streptavidin-coated magnetic beads (GE Healthcare), which were pre-centrifuged to allow the use of relatively small beads (~300 nm), were introduced into the flow chamber. Unbound beads were washed with more than 50 μL of observation buffer containing the prescribed concentrations of ATP and/or ADP/$P_i$. Rotations of the magnetic beads were observed using a phase-contrast microscope (IX-70, Olympus) with a 100× objective lens at a recording rate of 30 fps. The rotation assay was performed at 23 ± 2 °C. The ATP-regenerating system (0.2 mg/mL pyruvate kinase and 2.5 mM phosphoenolpyruvate) was added to the solution mixture, except when ADP was used.

To observe the $IF_1$ inhibitory state, we identified rotating molecules among the particles attached to the coverslip in the $IF_1$-free solution. Next, a solution containing $IF_1$ in ATP solution or ATP synthesis buffer (100 μM ATP, 100 μM ADP, and 1 mM $P_i$) was infused into the flow chamber. After rotations for several tens of seconds, all $bMF_1$ molecules stopped rotation (Fig. 2a).

### Manipulation with magnetic tweezers

Magnetic tweezers, consisting of two sets of electromagnets, were equipped onto the microscope stage and controlled by custom-made

software[20,58]. Rotations were imaged at 30 fps using a progressive-scan camera (FC300M, Takex, Kyoto, Japan), which allowed real-time manipulation with magnetic tweezers. Movies were stored on a computer as AVI files and analyzed using custom-made software.

## Kinetic analysis of ADP inhibition

Rotations in the $IF_1$-free solution were recorded for more than 10 min per molecule (Supplementary Fig. 1). Under these conditions, ATP binding occurs for less than 1 ms and cannot be detected as a pause of rotations with magnetic beads at a recording rate of 30 fps. However, $bMF_1$ showed frequent transient pauses; we collected the data from all pauses longer than 1 s[31], and rotary traces between pauses longer than 1 s were defined as rotations. Distributions of pausing times before the onset of rotations were fitted by a double exponential function, $y = N_{sp}\exp(-t/\tau_{sp}) + N_{lp}\exp(-t/\tau_{lp})$, where $N_{sp}$ and $N_{lp}$ are constants, and $\tau_{sp}$ and $\tau_{lp}$ are time constants for the short and long pause, respectively. A relatively short pause (sp) corresponds to decelerated $P_i$ release, as previously reported[31,70]. The long pause (lp) corresponds to the ADP-inhibited form. These results showed that the time scale for ADP inhibition was 10-30 s under the various conditions tested. Distributions of rotating times before lapsing into the pauses were fitted by a single exponential function, $y = N_{rot}\exp(-t/\tau_{rot})$, where $N_{rot}$ is a constant and $\tau_{rot}$ is a time constant for rotations. In Supplementary Fig. 1, the histograms include most of the total data; the remaining data are not shown in the figure for clarity. All data are provided in the Source Data.

## Analysis of $IF_1$ inhibition

The experimental procedure was described in the main text (see also Fig. 2a). Rotating time (Supplementary Figs. 2, 3, 6, 10, and 12) was defined as the period of time from the completion of solution exchange until when the molecules fell into $IF_1$ inhibition. Probability plots versus rotating time were fitted with a single-exponential decay function to estimate the mean rotation time, $\tau_{rot}^{IF_1}$. The figures include most of the total data; the remaining data are not shown in the figure for clarity. All data are provided in the Source Data. A *pause* was defined as a pause state for more than 1 s. Except for $IF_1(\Delta1-19)$ and $IF_1(\Delta1-22)$, most of the pauses showed extremely long pausing states of more than 480 s, corresponding to $IF_1$ inhibition. A few traces showed relatively short pausing states of ~20 s. They were attributable to ADP inhibition because the timescales of these pauses were mostly identical to the values estimated from the $IF_1$-free solution experiment (see Supplementary Fig. 1). All of them spontaneously recovered active rotation and eventually fell into $IF_1$ inhibition without exception. For $IF_1(\Delta1-19)$ and $IF_1(\Delta1-22)$, rotations were frequently recovered without manipulation, suggesting a reversible inhibition/dissociation mechanism. The histogram of pausing time was fitted by single-exponential decay function for $IF_1(\Delta1-19)$ to estimate the mean duration time, $\tau_{pause}^{IF_1}$, 254 s. For $IF_1(\Delta1-22)$ analysis, the double-exponential decay function was applied to fit the histogram, estimating $\tau_{lp}^{IF_1}$ and $\tau_{sp}^{IF_1}$. Although the underlying molecular mechanism corresponding to $\tau_{sp}^{IF_1}$ is unclear, we selected $\tau_{lp}^{IF_1}$ for $IF_1(\Delta1-22)$ inhibitory states.

## Reporting summary

Further information on research design is available in the Nature Portfolio Reporting Summary linked to this article.

# Data availability

The structural information used for Figs. 1b, 5a, b, and 6b is accessible in PDB accession number 2v7q [https://doi.org/10.2210/pdb2v7q/pdb]. Source data are provided with this paper.

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

## Acknowledgements

We thank S. Mori (The University of Tokyo) for technical advice, T. Sagawa (The University of Tokyo) for critical discussions regarding theoretical aspects of Maxwell's demon concept, and all members of the Noji Laboratory for their valuable comments. The computation was partially performed using Research Center for Computational Science, Okazaki, Japan (Project: 22-IMS-C189). This work was supported in part by a Grant-in-Aid for Scientific Research on Innovation Areas (JP19H05380 and JP21H00388 to H.U.), a Grant-in-Aid for Scientific Research (S; JP19H05624 to H.N.) from the Japan Society for the Promotion of Science, and a Research Grant from Human Science Frontier Program (Ref. No: RGP0054/2020 to H.N.).

## Author contributions

R.K., H.U., and H.N. designed the research; R.K. conducted all experiments and analyses; K.O. performed molecular dynamic simulation; R.K. and H.N. wrote the paper with support from H.U. and K.O..

## Competing interests

The authors declare no competing interests.
