## [Peer review file · Nature Communications]

REVIEWER COMMENTS

Reviewer #1 (Remarks to the Author):

The manuscript "Molecular mechanism on forcible ejection of ATPase inhibitory factor 1 from mitochondrial ATP synthase" by Kobayashi et al. reports new and interesting findings concerning the interactions of regulatory protein IF1 and F1-subcomplex of bovine mitochondrial ATP synthase. The authors used proteins heterologously expressed in *Escherichia coli*. The ATPase activity of F1-subcomplex was measured both by biochemical methods and by single-molecule assay when it was immobilized on a glass and a bead was attached to subunit gamma, so that in response to addition of ATP the authors could directly observe bead rotation. The latter method is a very powerful technique that allows to analyze enzyme activity and its regulation in molecular details.

The authors found that IF1 is a very efficient inhibitor of F1 ATPase activity. At temperature of 23°C it bound to F1 with a time constant of 20 s at ATP concentration of 1 mM. This finding suggests that IF1 might promptly block ATP hydrolysis by FoF1 in mitochondria under de-energized conditions. The authors also demonstrated that IF1-inhibited F1 can be re-activated by forced rotation of subunit gamma in the ATP synthesis direction, but not in the hydrolysis direction. This is a new and important finding; previous studies of two other common regulatory mechanisms of bacterial F1 (ADP-inhibition and subunit epsilon) demonstrated activation in response to forced rotation in both directions. Finally, by using mutated variants of IF1 (with N-terminal truncations of 7, 12, 19, and 22 amino acid residues) the authors established that the interaction of N-terminal region of IF1 with subunit gamma is important for stabilization of the IF1-inhibited state of F1.

The manuscript is well-written in good English, the logic is clear, the methods are described in sufficient details, and the results support the conclusions. I think the manuscript can be published in *Nature Communications* after solving some issues described below.

1. All the experiments were done at 23°C, while the physiological temperature for a cow is about 37°C; at 23°C body temperature a cow is kind of dead. It is clear that repeating the single-molecule experiments at 37°C is not an option (lots of work + technically challenging), but it is possible to repeat the biochemical activity measurements and the effects of IF1 at 37°C. ATP synthase regulatory mechanisms are known to be temperature-dependent, so demonstrating that 23°C and 37°C experiments provide qualitatively similar results is important.

2. It was shown for *Bacillus PS3* F1 (Ref. 34) that activation by forced rotation from ADP-inhibited state and from epsilon-inhibited state requires more energy than ATP synthesis. Is it possible to estimate, whether this is also the case with IF1 inhibition of bovine F1? This is a very important issue, because it affects the pattern of activation in whole mitochondria, where many copies of ATP synthase share the same coupling membrane. It might also explain why some research groups observed the decrease in ATP synthesis rate upon IF1 association to FoF1.

3. It is stated in the text (lines 179-181) that some molecules were excluded from the analysis. Since single-molecule assays provide the researcher with an option to choose "good" molecules and ignore

"bad" ones, it would be nice to include a table (in Supporting materials) that provides some statistics, e.g. how many molecules were in the viewfield of the microscope, how many of them were totally inactive, how many of the active ones were later rejected from the analysis.

4. The authors provide time constants for rotation phase and inhibition phase. However, for a biochemist it is more important to know, what percentage of time F1 was active, and what percentage of time it was inactive. (It can be calculated from the time constants, but it is not quite simple). I suggest that the information on the fractions of time that the enzyme was active/inactive should be added to the results section

Minor points:

Line 34 - "counterclockwise" is ambiguous, until the reader knows from which side we are looking

Line 40 - "rotated" is kind of confusing here, because rotation is generally associated with the rotor rotation. May be use the term "swing-out motion" as in line 278?

Line 46 - ATP synthesis is the primary role of FoF1 in mitochondria, but not in many bacteria. Better to specify it.

Line 53 - I have not seen any data direct on the effect of pmf on IF1 association. In bovine mitochondria pH is a crucial factor that controls IF1 association, and I strongly recommend to add this fact to the Introduction. Acidification of the matrix usually results from anoxia and is accompanied by the drop in pmf, but it is difficult to prove the causal link between pmf and IF1 association.

Line 140 - what was the concentration of ATP?

Line 268 - the authors demonstrated that Pi also increases the probability of IF1 release upon gamma rotation, so ADP is not critically important for that. The discussion should be corrected accordingly.

Lines 342-347 - I think that the data on alanine substitutions are interesting and informative; may be include them in the results section?

Finally, citing some recent reviews on the regulatory mechanisms of ATP synthase might also improve the manuscript.

A PDF file with most of the comments above and with a few typos marked is also attached.

Conclusion: it is a well presented, laborious, elegant, and accurate experimental work. It will certainly be of interest to researchers in the fields of biochemistry, biophysics, bioenergetics, cell biology, and physiology.

Reviewer #2 (Remarks to the Author):

The manuscript demonstrates the mechanism of release of the inhibitor IF1 from ATP-synthase, using magnetic tweezers to mimic and assist ATP synthesis. Technically it is exemplary, the conclusions are demonstrated beautifully, clearly, and beyond doubt using a method that while not novel remains extremely challenging. I am not qualified to judge the significance of the result to the field of inhibitory control of ATP synthase. From the perspective of single-molecule biology, it is of great interest to a general reader as an example of the use of single molecule methods to reveal mechanistic details in biological molecular machines.

Reviewer #3 (Remarks to the Author):

This study reports single-molecule manipulation experiments with immobilized bovine F1 linked to magnetic beads to elucidate the molecular mechanism of F1 inhibition by its natural inhibitory peptide IF1. The results showed that IF1-inhibited F1 was efficiently activated only when F1 was forcibly rotated (with magnetic tweezers) in the clockwise (ATP synthesis) direction, but not in the counter-clockwise direction (ATP hydrolysis), and the activation was enhanced in the presence of ADP and Pi. These observations suggest that the principal mechanism for the pmf-induced activation of IF1-inhibited FoF1 (in membranes) is the ejection of IF1 from F1 by pmf-driven Fo motor rotation. The results obtained with IF1 mutants strongly support a model in which the inhibitory effect of IF1 is mainly due the prevention of subunit beta conformational changes rather than changes in the interactions of IF1 with subunit gamma. These are major findings that provide novel and important insights into the mechanisms that regulate the activity of F1FO complexes. The presentation of the work is excellent.

I have only a few minor points:

-In the introduction (lines 52, 53), the authors may want to include a previous reference (Lefebvre-Legendre et al. Mol. Microbiol 2003) showing that F1-mediated ATP hydrolysis is essential for cell viability, for instance in cells of *Saccharomyces cerevisiae* lacking functional mitochondrial DNA (and hence the FO) where this activity is absolutely needed to maintain a sufficient mitochondrial membrane potential by the ADP/ATP translocase. In these pure ATP hydrolysis conditions, inhibition of F1 by IF1 would be lethal rising the possibility that IF1 does not necessarily bind to the F1 when it cannot dock to FO?

-The formulation “activation of IF1 inhibition” used all along the manuscript to indicate that F1 inhibition by IF1 is released is a bit confusing because it suggests instead that F1 becomes inhibited by IF1.

-line 79. Remove the ‘it’ before F1.

**Molecular mechanism on forcible ejection of ATPase inhibitory factor 1
from mitochondrial ATP synthase**

Ryohei Kobayashi^{a,b}, Hiroshi Ueno^a, Kei-ichi Okazaki^{b,c}, Hiroyuki Noji^{a,*}

^a *Department of Applied Chemistry, Graduate School of Engineering, The University of Tokyo, Hongo, Tokyo 113-8656, Japan*

^b *Research Center for Computational Science, Institute for Molecular Science, Okazaki, Aichi 444-8585, Japan*

^c *The Graduate University for Advanced Studies, Sokendai, Okazaki, Aichi 444-8585, Japan*

*Corresponding author: Hiroyuki Noji

Phone number: +81-3-5841-7252

Email: hnoji@g.ecc.u-tokyo.ac.jp

**Abstract**

IF₁ is a natural inhibitor protein for mitochondrial F₀F₁ ATP synthase that blocks catalysis and rotation
of the F₁ by deeply inserting the N-terminal helices into F₁. A unique feature of IF₁ is the condition-
dependent inhibition; although IF₁ inhibits the ATP hydrolysis of F₁, IF₁ inhibition is relieved under
ATP synthesis conditions. To elucidate the condition-dependent inhibition mechanism, we performed
single-molecule manipulation experiments on IF₁-inhibited *bovine* mitochondrial F₁ (*bMF*₁). The
results showed that IF₁-inhibited F₁ was efficiently activated only when F₁ was rotated in the clockwise
(ATP synthesis) direction, but not in the counter-clockwise direction. The observed rotational-
direction-dependent activation explains the condition-dependent mechanism of IF₁ inhibition.
Investigation of mutant IF₁ with the N-terminal truncation showed that the interaction with the γ
subunit at the N-terminal regions is crucial for rotational-direction-dependent ejection, and the middle
long helix is responsible for the inhibition of F₁.

**Keywords:**

ATPase inhibitory factor 1, IF₁, F₁-ATPase, Single-molecule manipulation, ATP synthase, F₀F₁

Introduction

[revised manuscript text omitted]

solution. Thus, it is evident that reactivation of IF₁ inhibition occurs under ATP synthesis conditions.
With IF₁(WT), the probability of activation reached 100% when stalled at -320° for 5 s. This is
sufficiently high to explain the *pmf*-induced activation of the IF₁-inhibited F_oF₁⁵⁰⁻⁵⁵. Thus, the
principal mechanism for the *pmf*-induced activation of IF₁-inhibited F_oF₁ is the ejection of IF₁ from F₁
by forcible rotation powered by *pmf*-driven F_o motor in the F_oF₁ complex. The requirement of A_{DP}
and P_i for efficient activation indicates that forcible clockwise rotation should be coupled with the
ATP synthesis reaction for IF₁ ejection. As the presence of substrates enhances the cooperative nature
of F₁, IF₁ may likely be ejected through a concerted conformational transition of the α and β subunits
coupled with γ subunit rotation.

We observed a significant increase in the probability of activation when the IF₁-inhibited
*b*MF₁ was rotated over -200°. The crystal structure of *b*MF₁-IF₁ complex show that IF₁ binds to β_{
[revised manuscript text omitted]

exponential decay was not observed in the time course of IF₁(Δ1-22) inhibition (Fig. S8G), curve
fitting using (Eq. 1) were not performed.

If the reaction was irreversible, as shown in IF₁ (WT), IF₁ (Δ1–7), and IF₁ (Δ1–12), the following
 reaction scheme may represent sequential IF₁ inhibition:

where $F_1 \cdot IF_1^{lock}$ represents the inactive state, and $F_1 \cdot IF_1$ is the intermediate state that is still
 catalytically active. k_{on} and k_{off} represent the rate constants of association and dissociation,
 respectively. k_{lock} is the rate constant of isomerization to the locked state. According to (Scheme 1),
 $k_{inhibition}^{app}$ can be expressed as follows:

$$424 \quad k_{inhibition}^{app} = \frac{k_{lock}[IF_1]}{K_M^{IF_1} + [IF_1]} \quad (\text{Eq. 2})$$

$$425 \quad K_M^{IF_1} \equiv \frac{k_{off} + k_{lock}}{k_{on}} \quad (\text{Eq. 3})$$

which shows a typical hyperbolic relationship.

*Rotation assay of bMF₁*

For the rotation assay, bMF₁ was immobilized onto the glass surface, and magnetic beads
 were attached to the two cysteine residues of the γ subunit (γ A99C and γ S191C) through a biotin-
 avidin interaction. The protocol was as follows²⁰. A flow chamber (~5 μ L) was prepared with two
 coverslips (18 \times 18 mm² at the top, and 24 \times 32 mm² at the bottom; Matsunami Glass) with double-
 sided tape (7602 #25, Teraoka) as a spacer. Purified bMF₁ (~500 pM) in observation buffer (50 mM
 HEPES-KOH (pH 7.5), 50 mM KCl, and 2 mM MgCl₂) was gently introduced into the flow chamber
 and incubated for 5–10 min. After washing unbound bMF₁ with more than 50 μ L of observation buffer
 containing ~10 mg/mL BSA, streptavidin-coated magnetic beads (GE Healthcare), which were pre-
 centrifuged to enable the use of relatively small beads (~300 nm), were introduced into the flow
 chamber. Unbound beads were washed with more than 50 μ L of observation buffer containing the
 prescribed concentrations of ATP and/or ADP/P_i. Rotations of the magnetic beads were observed
 using a phase-contrast microscope (IX-70, Olympus) with a 100 \times objective lens at a recording rate of
 30 fps. The rotation assay was performed at 23 \pm 2 $^{\circ}$ C. The ATP-regenerating system (0.2 mg/mL
 pyruvate kinase and 2.5 mM phosphoenolpyruvate) was added to the solution mixture unless ADP
 was used.

To observe the IF₁ inhibitory state, we identified rotating molecules among the particles
 attached to the coverslip in the IF₁-free solution. Next, a solution containing IF₁ and ATP/ATP
 synthesis (100 μ M ATP, 100 μ M ADP, and 1 mM P_i) was infused into the flow chamber. After rotation,
 bMF₁ molecules eventually stopped rotation (Fig. 2A).

*Manipulation with magnetic tweezers*

Magnetic tweezers, comprising two sets of electromagnets, were equipped onto the
microscope stage and controlled by custom-made software^{20,56}. Rotations were imaged at 30 fps using
a progressive-scan camera (FC300M, Takex, Kyoto, Japan), which enabled real-time manipulation
using magnetic tweezers. Movies were stored on a computer as AVI files and analyzed using custom-
made software.

*Kinetic analysis of ADP inhibition*

Rotations in the IF₁-free solution were recorded for > 10 min per molecule (Fig. S1). Under
these conditions, ATP binding occurs for less than 1 ms, and therefore, cannot be detected as a pause
of rotations with magnetic beads at a recording rate of 30 fps. However, bMF₁ showed frequent
transient pauses; we collected the data from all pauses longer than 1 s²⁹ and rotary traces between
pauses longer than 1 s were defined as rotations. The distribution of pausing times before starting
rotations was fitted by a double exponential function, $y = N_{sp} \exp(-t/\tau_{sp}) + N_{lp} \exp(-t/\tau_{lp})$,
where N_{sp} and N_{lp} are constants, and τ_{sp} and τ_{lp} are time constants for the short and long pause,
respectively. A relatively short pause (sp) corresponds to decelerated P_i release, as previously reported
464^{29,67}. Long pause (lp) corresponds to the ADP-inhibited form. These results showed that the timescale
for ADP inhibition was 10-30 s under the various conditions tested. The distribution of rotating times
before lapsing pauses was fitted by a single exponential function, $y = N_{rot} \exp(-t/\tau_{rot})$, where
N_{rot} is a constant and τ_{rot} is a time constant for rotations.

*Analysis of IF₁ inhibition*

The experimental procedure was described in the main text (see also Fig. 2A). Rotating time
(Fig. S2, S3, S6, S10, and S12) was defined as the period of time from the completion of solution
exchange until when the molecules fell into IF₁ inhibition. Probability plots versus rotating time was
fitted by a single-exponential decay function to estimate the mean rotation time, $\tau_{rot}^{IF_1}$. Pauses were
defined as that longer than 1 s, observed in the solution containing IF₁. Except IF₁(Δ1-19) and IF₁(Δ1-
22), most of the pauses showed extremely long pausing state over 480 s, corresponding to IF₁
inhibition. A few traces showed relatively short pausing state of ~20 s. They were attributable to ADP
inhibition because the timescales of these pauses were mostly identical to that estimated from IF₁-free
solution experiment (see Fig. S1). All of them spontaneously recovered active rotation and eventually
fell into IF₁ inhibition without exception. For IF₁(Δ1-19) and IF₁(Δ1-22), rotations were frequently
recovered without manipulation indicating a reversible inhibition/dissociation mechanism. The
histogram of pausing time was fitted by single-exponential decay function for IF₁(Δ1-19) to estimate
the mean duration time, $\tau_{pause}^{IF_1}$, 254 s. For IF₁(Δ1-22) analysis, the double-exponential decay
function was applied to fit the histogram, estimating $\tau_{lp}^{IF_1}$ and $\tau_{sp}^{IF_1}$. Although the underlying

molecular mechanism corresponding to $\tau_{sp}^{IF_1}$ is unclear, we selected $\tau_{tp}^{IF_1}$ for IF₁(Δ1–22) inhibitory
states.

*Selection of molecules in the manipulation experiments*

To discriminate IF₁ inhibition from ADP inhibition, we adopted the following method in the
manipulation experiments. For forcible 360° manipulation experiment (Fig. 3, 5 and Fig. S5, S12), we
waited for 480 s after confirming a pause, because IF₁ inhibition was irreversible and did not recover
rotation within the observation period (Fig. S2A) whereas ADP inhibition was reversible and
spontaneously recovered rotation. In the “stall-and-release” experiment, we selectively analyzed the
molecule that showed activation by 360° clockwise rotation. Thus, the maximum probability shown
in the figures (e.g. Fig. 4C) was 100%, whereas around 60% in the forcible 360° manipulation where
all molecules are subject to analysis.

It is noted that the difference between IF₁ inhibition and ADP inhibition appeared in the
activation probability by counterclockwise manipulation, as well as the length of the pausing time.
Pauses observed in IF₁-free solution can be almost completely activated by the stall at +80° in the
counterclockwise direction for 2 or 5 s (Fig. S7, triangles). By contrast, IF₁ inhibition, which was
defined as the pauses longer than 480 s under saturating IF₁ conditions, was not activated by the same
manipulation (Fig. S7, squares). Thus, IF₁-inhibited state was confirmed by the following way; after
re-inactivation by IF₁ in solution, we stalled the targeted molecules at +80° in the clockwise direction
before each trial. We analyzed the molecule that did not show reactivation by the abovementioned
manipulation.

Through repeated manipulations with magnetic tweezers, molecules sometimes showed
unusual fluctuations. This may reflect the unstable states of the molecules or their detachment from
the coverslip. Such molecules cannot be reactivated by forcible 360° clockwise rotation.
Contamination of these molecules in the analysis lowers the probability of reactivation. We
concluded that reliable data cannot be obtained if such disordered molecules are subjected to further
experiments. For molecules that no longer resumed active rotation after 360° clockwise rotation, we
completed the manipulation of the molecules and searched for a new molecule.

*Probability of spontaneous reactivation during manipulation*

Unlike IF₁(WT), IF₁(Δ1–7) and IF₁(Δ1–12), IF₁(Δ1–19) and IF₁(Δ1–22) did not stop rotations
of bMF₁ completely. Considering the inherent pausing time of these two mutants, 250 s and 60 s,
respectively (Fig. S10), we selectively manipulated pausing states that last for longer than 100 s for
IF₁(Δ1–19) and 60 s IF₁(Δ1–22).

Inhibitory states by IF₁(Δ1–22) were reactivated by manipulation of 360° at the rate of 0.5
rps, that is, 2 s per manipulation. To exclude the possibility that rotations resumed irrespective of

manipulation for 2 s, we performed the following calculation. The mean pausing time by IF₁(Δ1–22)
was 60 s, whereas short pauses, lasting only 2 s on average, were also observed. As mentioned above,
we waited for 60 s to selectively manipulate the long pauses after confirming the pauses. Thus, the
impact of short pauses on this analysis was negligible. The probability that these pausing states end in
2 s is calculated using the following equation:

$$100 \times \frac{\int_0^2 \exp\left(-\frac{t}{60}\right) dt}{\int_0^\infty \exp\left(-\frac{t}{60}\right) dt} = 3\%$$

Therefore, the probability of simultaneous reactivation was so small that we concluded that the
reactivation of rotation was invoked by the manipulation itself.

Fig. 1. Outline of this work.

(A) Rotation scheme of *bMF*₁. Circles represent the catalytic states of bound nucleotides at β subunit. ATP* in the circles at 80°, 200°, and 320° represents the catalytically active state where the bound ATP is to be hydrolyzed. Arrows represent the rotary angles of γ subunit. 0° is defined as the position of the γ subunit where the blue β subunit binds to ATP. *Short pauses* at 10°-20°, 130°-140°, and 250°-260°, observed in our previous paper²⁰, are omitted from this figure for clarity.

(B) Crystal structure of *bMF*₁ with IF₁ bound to the $\alpha\beta_{DP}$ subunit (PDB: 2v7q). α_{DP} , β_{DP} , γ and IF₁ are colored by dark red, pink blue, and green, respectively. β_{DP} subunit is omitted for clarity in the enlarged illustration.

(C) An illustration of the single-molecule rotation assay system of *bMF*₁. The stator $\alpha_3\beta_3$ -ring is immobilized onto a glass surface. A magnetic bead ($\varphi \sim 300$ nm) is attached to the rotor γ subunit as a rotation probe through biotin-streptavidin interaction. Magnetic tweezers, comprising of two sets of electromagnets, were equipped onto the sample stage of the microscope.

**Fig. 2. Single-molecule analysis of IF₁-inhibited bMF₁.**

(A) Experimental method. After observing a rotating bMF₁ molecule in IF₁-free solution, we
 introduced a mixture solution with IF₁ into the reaction chamber. The bMF₁ continued to rotate
 for a while but finally stops rotation.

(B) Typical time-courses of bMF₁ in the presence of 3 μM IF₁ and 1 mM ATP. For more detailed
 analysis, see Fig. S2.

(C) Stall positions of IF₁ inhibition. (Left) An example of the IF₁ inhibited pauses. After observing
 the ATP-binding waiting dwell at 100 nM ATP, 5 μM IF₁ with 100 nM ATP was introduced into
 the reaction chamber. Blue data points represent stalls of IF₁ inhibition. (Right) The angular
 distance ($\Delta\theta$) of IF₁-inhibited from the left-side ATP-binding waiting dwell (pink) ($N = 29$ pauses).
 Values represent mean \pm SD estimated from Gaussian fitting of the datapoints.

**Fig. 3. Single-molecule manipulation of IF₁-inhibited *b*MF₁.**

(A) Schematic images of manipulation procedure. When *b*MF₁ was stalled by IF₁, the magnetic
 tweezers were turned on to stall *b*MF₁ to rotate one clockwise or counterclockwise revolution at
 the rate of 0.5 rps. After manipulation, released *b*MF₁ either resumed its rotation (ON) or not
 (OFF). These behaviors indicate whether IF₁ is dissociated from *b*MF₁ under stalling time or not,
 respectively.

(B) Reactivation probability of IF₁-inhibited *b*MF₁ after manipulation. The reactivation probability
 (P_{ON}) was defined as that of an ON event against total molecules. "Hyd" and "Syn" represent the
 direction for hydrolysis (counterclockwise) and synthesis (clockwise), respectively. Error bars
 represent SD, given as $\sqrt{P_{ON}(100 - P_{ON})/N}$, where N is the number of total molecules ($N =$
 19-26 molecules).

**Fig. 4. The stall-and-release experiment of IF_1 -stalled bMF_1 .**

(A) Schematic images of manipulation procedure in the stall-and-release experiment. When bMF_1
 was stalled by IF_1 , the magnetic tweezers were turned on to stall bMF_1 at the targeted angle. After
 the set time elapsed, the magnetic tweezers were turned off, and the molecule went back to the
 initial angle. Released bMF_1 either resumes its rotation (ON) or stays at the initial position (OFF).
 These behaviors indicate whether IF_1 is dissociated from bMF_1 under stalling time or not,
 respectively.

(B) A representative time course of stall-and-release experiment under 100 μ M ATP, 100 μ M ADP, 1
 mM P_i in the presence of 3 μ M IF_1 . In this figure, the stall time for both trials (blue) was 5 s and
 the stall angle are -26 $^\circ$ and -168 $^\circ$, respectively.

(C) Angle dependence of reactivation probability under 100 μ M ATP, 100 μ M ADP, 1 mM P_i in the
 presence of 3 μ M IF_1 . Each data point was obtained from 15 to 47 trials using 4 to 14 molecules.
 Counterclockwise rotation (blue) and clockwise rotation (orange) is defined as positive and
 negative direction, respectively. Colors on the plots represent the stall time of 0.5 s (red), 2 s
 (grey) and 5 s (blue), respectively. Error bars represent SD, given as $\sqrt{P_{ON}(100 - P_{ON})/N}$,
 where N is the number of total trials in each datapoint.

Fig. 5. Analysis of mutant IF₁s with N-terminal truncation

(A) Enlarged illustration of interactions between IF₁ and γ subunit in the *b*MF₁-IF₁ complex (PDB: 2v7q). S11 and F22 in IF₁ (green) can interact with N15 and I16 in γ subunit (blue), respectively.

(B) Details of IF₁ structure (PDB: 2v7q), where residues 8-50 are resolved. Residues 1-7 are not resolved. Residues 8-13 are resolved and form an extended structure. Residues 14-18 and 21-50 form short and long helix, respectively, linked by glycine loop in residues 19-20.

(C) Sequence of bovine IF₁¹⁻⁶⁰ and definition of N-terminal truncated mutants.

(D) & (E) Reactivation probability of inhibited *b*MF₁ by mutant IF₁s after (D) counterclockwise and (E) clockwise manipulation. Experiments were conducted under 100 μ M ATP, 100 μ M ADP, and 1 mM P_i. The reactivation probability (P_{ON}) was defined as that of an ON event against total trials.

The data for IF₁(WT) is the same as Fig. 3B. Error bars represent SD, given as

$\sqrt{P_{ON} (100 - P_{ON})/N}$, where N is the number of total molecules ($N=11-25$ molecules).

**Fig. 6. The proposed mechanisms presented in this study**

(A) Coupling scheme of IF₁ dissociation with the rotary mechanism in *b*MF₁. A part of the rotation
 scheme of *b*MF₁ (Fig. 1A) is described with only the essential elements. The green oval represents
 IF₁. The green line in the scheme represents the IF₁ binding angle ranging with -40°-0°. The blue
 line represents the angular difference (0°-200°) required for the conformational transition of the
 blue β subunit from β_E to β_{DP} via β_{TP} . The black line represents the angle distance for IF₁
 dissociation, estimated in the stall-and-release experiment.

(B) The schematic image of the role of IF₁ where residues 8-50 are resolved (PDB: 2v7q). In addition
 to residues 1-19 (purple), residues 20-22 (orange) are critical for the unidirectionality. Residues
 from 23 (green) are required for inhibition.

**Reference**

- 1. Noji, H., Ueno, H. & McMillan, D. G. G. Catalytic robustness and torque generation of the
F1-ATPase. *Biophys Rev* **9**, 103–118 (2017).
- 2. Junge, W., Lill, H. & Engelbrecht, S. ATP synthase: An electrochemical transducer with
rotatory mechanics. *Trends Biochem Sci* **22**, 420–423 (1997).
- 3. Yoshida, M., Muneyuki, E. & Hisabori, T. ATP synthase - A marvellous rotary engine of the
cell. *Nat Rev Mol Cell Biol* **2**, 669–677 (2001).
- 4. Walker, J. E. E. The ATP synthase: The understood, the uncertain and the unknown.
*Biochem Soc Trans* **41**, 1–16 (2013).
- 5. Kühlbrandt, W. Structure and Mechanisms of F-Type ATP Synthases. (2019)
doi:10.1146/annurev-biochem-013118.
- 6. Xu, T., Pagadala, V. & Mueller, D. M. Understanding structure, function, and mutations in
the mitochondrial ATP synthase. *Microbial Cell* **2**, 105–125 (2015).
- 7. Noji, H., Ueno, H. & Kobayashi, R. Correlation between the numbers of rotation steps in the
ATPase and proton-conducting domains of F- and V-ATPases. *Biophys Rev* 303–307 (2020)
doi:10.1007/s12551-020-00668-7.
- 8. Abrahams, J. P., Leslie, A. G. W., Lutter, R. & Walker, J. E. Structure at 2.8 Å resolution of
F1-ATPase from bovine heart mitochondria. *Nature* **370**, 621–628 (1994).
- 9. Cingolani, G. & Duncan, T. M. Structure of the ATP synthase catalytic complex (F1) from
*Escherichia coli* in an autoinhibited conformation. *Nat Struct Mol Biol* **18**, 701–707 (2011).
- 10. Kabaleeswaran, V., Puri, N., Walker, J. E., Leslie, A. G. W. & Mueller, D. M. Novel features
of the rotary catalytic mechanism revealed in the structure of yeast F1 ATPase. *EMBO*
*Journal* **25**, 5433–5442 (2006).
- 11. Morales-Rios, E., Montgomery, M. G., Leslie, A. G. W. & Walker, J. E. Structure of ATP
synthase from *Paracoccus denitrificans* determined by X-ray crystallography at 4.0 Å
resolution. *Proceedings of the National Academy of Sciences* **112**, 13231–13236 (2015).
- 12. Sobti, M., Ueno, H., Noji, H. & Stewart, A. G. The six steps of the complete F1-ATPase
rotary catalytic cycle. *Nat Commun* **12**, (2021).
- 13. Bowler, M. W., Montgomery, M. G., Leslie, A. G. W. & Walker, J. E. Ground State
Structure of F1-ATPase from Bovine Heart Mitochondria at 1.9 Å Resolution. *Journal of*
*Biological Chemistry* **282**, 14238–14242 (2007).
- 14. Noji, H. & Ueno, H. How Does F1-ATPase Generate Torque?: Analysis From Cryo-Electron
Microscopy and Rotational Catalysis of Thermophilic F1. *Front Microbiol* **13**, (2022).
- 15. Börsch, M. & Duncan, T. M. Spotlighting motors and controls of single FoF1-ATP synthase.
*Biochem Soc Trans* **41**, 1219–1226 (2013).

- 16. Sielaff, H. & Börsch, M. Twisting and subunit rotation in single FOF1-ATP synthase.
*Philosophical Transactions of the Royal Society B: Biological Sciences* vol. 368 Preprint at
<https://doi.org/10.1098/rstb.2012.0024> (2013).

[revised manuscript text omitted]

*Chemistry* **291**, 538–546 (2016).
- 61. Zarco-Zavala, M. *et al.* The $3 \times 120^\circ$ rotary mechanism of *Paracoccus denitrificans* F1-
ATPase is different from that of the bacterial and mitochondrial F1-ATPases. *Proceedings of*
*the National Academy of Sciences* 202003163 (2020) doi:10.1073/pnas.2003163117.
- 62. Chui-Fann, W. & Gerhard, G. The Unique C-Terminal Extension of Mycobacterial F-ATP
Synthase Subunit α Is the Major Contributor to Its Latent ATP Hydrolysis Activity.
*Antimicrob Agents Chemother* **64**, e01568-20 (2020).
- 63. Harikishore, A. *et al.* Targeting Mycobacterial F-ATP Synthase C-Terminal α Subunit
Interaction Motif on Rotary Subunit γ . *Antibiotics* **10**, 1456 (2021).
- 64. Montgomery, M. G., Petri, J., Spikes, T. E. & Walker, J. E. Structure of the ATP synthase
from *Mycobacterium smegmatis* provides targets for treating tuberculosis. *Proceedings of*
*the National Academy of Sciences* **118**, e2111899118 (2021).
- 65. Andrianaivomananjaona, T., Moune-Dimala, M., Herga, S., David, V. & Haraux, F. How the
N-terminal extremity of *Saccharomyces cerevisiae* IF1 interacts with ATP synthase: A
kinetic approach. *Biochimica et Biophysica Acta (BBA) - Bioenergetics* **1807**, 197–204
(2011).
- 66. Ma, J. *et al.* A Dynamic Analysis of the Rotation Mechanism for Conformational Change in
F1-ATPase. *Structure* **10**, 921–931 (2002).
- 67. Watanabe, R., Hayashi, K., Ueno, H. & Noji, H. Catalysis-Enhancement via Rotary
Fluctuation of F1-ATPase. *Biophys J* **105**, 2385–2391 (2013).

Response to Reviewer comments [NCOMMS-22-41840]

We sincerely appreciate the Reviewers for their careful reading and constructive comments. According to the comments, we have revised the manuscript. A step-by-step response is provided below (reviewer comments in *green* and our replies in black).

Reviewer #1

The manuscript "Molecular mechanism on forcible ejection of ATPase inhibitory factor 1 from mitochondrial ATP synthase" by Kobayashi et al. reports new and interesting findings concerning the interactions of regulatory protein IF1 and F1-subcomplex of bovine mitochondrial ATP synthase. The authors used proteins heterologally expressed in Escherichia coli. The ATPase activity of F1-subcomplex was measured both by biochemical methods and by single-molecule assay when it was immobilized on a glass and a bead was attached to subunit gamma, so that in response to addition of ATP the authors could directly observe bead rotation. The latter method is a very powerful technique that allows to analyze enzyme activity and its regulation in molecular details.

The authors found that IF1 is a very efficient inhibitor of F1 ATPase activity. At temperature of 23°C it bound to F1 with a time constant of 20 s at ATP concentration of 1 mM. This finding suggests that IF1 might promptly block ATP hydrolysis by FoF1 in mitochondria under de-energized conditions. The authors also demonstrated that IF1-inhibited F1 can be re-activated by forced rotation of subunit gamma in the ATP synthesis direction, but not in the hydrolysis direction. This is a new and important finding; previous studies of two other common regulatory mechanisms of bacterial F1 (ADP-inhibition and subunit epsilon) demonstrated activation in response to forced rotation in both directions. Finally, by using mutated variants of IF1 (with N-terminal truncations of 7, 12, 19, and 22 amino acid residues) the authors established that the interaction of N-terminal region of IF1 with subunit gamma is important for stabilization of the IF1-inhibited state of F1.

The manuscript is well-written in good English, the logic is clear, the methods are described in sufficient details, and the results support the conclusions. I think the manuscript can be published in Nature Communications after solving some issues described below.

We thank the reviewer for taking the time for reviewing and giving helpful comments. We revised the manuscript along the comments as below.

(1) *All the experiments were done at 23°C, while the physiological temperature for a cow is about 37°C; at 23°C body temperature a cow is kind of dead. It is clear that repeating the single-molecule experiments at 37°C is not an option (lots of work + technically challenging), but it is possible to repeat the biochemical activity measurements and the effects of IF1 at 37°C. ATP synthase regulatory mechanisms are known to be temperature-dependent, so demonstrating that 23°C and 37°C experiments provide qualitatively similar results is important.*

Along the comment, we conducted biochemical analysis of the representative mutant IF1s IF1(Δ 1-19) and IF1(Δ 1-22) and the wild-type, IF1(WT) to confirm that fundamental aspects are observed at more physiological condition, at 37°C. We show below the time course of inhibition

(left; 25°C, center; 37°C, in (A), (C), (E)) and the resulting residual ATPase activity (right, in (B), (D), (F)). The result at 37°C was essentially identical to that at 25°C, as seen in the original manuscript: IF₁(WT) showed complete inhibition with no residual activity, whereas IF₁(Δ1-19) and IF₁(Δ1-22) did not completely inhibit ATPase activity. We also found that the efficiency of IF₁ inhibition at 37°C was about five-fold lower than that at 25°C. This observation was well consistent with the previous finding that the chemical equilibrium between inhibition and dissociation in IF₁ kinetics is shifted to dissociation when temperature increases (Ferguson, *et al.*, *Biochem. J.* 1977).

In summary, our biochemical experiment suggests that the principle of IF₁ inhibition and dissociation is preserved between at room temperature and at physiological temperatures of mitochondria, 37°C, supporting the generality of the findings in the present study. We have revised the *Discussion* (line 330-338), providing the figure of this experimental result to Supplementary Fig. 13 and 14.

Figure. Biochemical analysis under 37°C.

(A, C, E) Time courses for IF₁-inhibited bMF₁ at 25°C and 37°C with the indicated concentrations of (A) IF₁(WT) (black/gray), (C) IF₁(Δ1-19) (green), (E) IF₁(Δ1-22) (purple). (B, D, F) Residual ATPase activity versus [IF₁] at the end of the measurement (350 s after IF₁ injection). Colored squares (black, green, purple) represent results at 37°C and gray circles represent results at 25°C (shown in Supplementary Fig. 8), respectively. Symbols and error bars represent the mean value and SD, respectively (*N* = 3 for each measurement).

- (2) *It was shown for Bacillus PS3 F1 (Ref. 34) that activation by forced rotation from ADP-inhibited state and from epsilon-inhibited state requires more energy than ATP synthesis. Is it possible to estimate, whether this is also the case with IF1 inhibition of bovine F1? This is a very important issue, because it affects the pattern of activation in whole mitochondria, where many copies of ATP synthase share the same coupling membrane. It might also explain why some research groups observed the decrease in ATP synthesis rate upon IF1 association to FoF1.*

This is an interesting and important question from mechanistic and energetic point of view. However, this study does not quantify the torque at which magnetic tweezers exert F_1 . Therefore, it is not possible to directly estimate the minimum membrane voltage for the injection of IF_1 . However, when bMF_1 generates torque of $40 \text{ pN} \cdot \text{nm}$ like other F_1 s, the minimum value for proton motive force as voltage to reverse F_1 is 196 mV, considering the proton stoichiometry of 8 protons/turn. In addition, the minimum energy for IF_1 ejection is assumed to be $140 \text{ pN} \cdot \text{nm}$ (84 kJ/mol). Although these estimations are tentative, we provide this discussion in the revised manuscript (line 295-300).

- (3) *It is stated in the text (lines 179-181) that some molecules were excluded from the analysis. Since single-molecule assays provide the researcher with an option to choose "good" molecules and ignore "bad" ones, it would be nice to include a table (in Supporting materials) that provides some statistics, e.g. how many molecules were in the view field of the microscope, how many of them were totally inactive, how many of the active ones were later rejected from the analysis.*

Along this comment, we have added a table about information in the single-molecule experiment to Supplementary Table 6.

- (4) *The authors provide time constants for rotation phase and inhibition phase. However, for a biochemist it is more important to know, what percentage of time F1 was active, and what percentage of time it was inactive. (It can be calculated from the time constants, but it is not quite simple). I suggest that the information on the fractions of time that the enzyme was active/inactive should be added to the results section.*

Thank you for your suggestions. Along this, we provide tables for comparison of active/inactive states to Supplementary Table 4 and 5. For biochemical measurements of IF_1 inhibition, readers can obtain similar information from the 'Residual ATPase Activity versus $[IF_1]$ ' shown in the Supplementary Figure 8 and 10.

● *Minor comment*

- (5) *Line 34 - "counterclockwise" is ambiguous, until the reader knows from which side we are looking*

Thank you for the advice. Along this comment, we added the description of 'when viewed from the membrane side' in the corresponding part.

- (6) *Line 40 - "rotated" is kind of confusing here, because rotation is generally associated with the rotor rotation. May be use the term "swing-out motion" as in line 278?*

We have revised the corresponding part along the suggestion.

- (7) *Line 46 - ATP synthesis is the primary role of FoF1 in mitochondria, but not in many bacteria. Better to specify it.*

Thank you for the constructive comment. Along this, we revised the manuscript to notify readers of this contention.

- (8) *Line 53 - I have not seen any data direct on the effect of pmf on IF1 association. In bovine mitochondria pH is a crucial factor that controls IF1 association, and I strongly recommend to add this fact to the Introduction. Acidification of the matrix usually results from anoxia and is accompanied by the drop in pmf, but it is difficult to prove the causal link between pmf and IF1 association.*

Thank you for your careful reading. Along this suggestion, we have revised the manuscript.

- (9) *Line 140 - what was the concentration of ATP?*

Along the comment, we added the description for the exact ATP concentration (1 mM) in the revised manuscript.

- (10) *Line 268 - the authors demonstrated that Pi also increases the probability of IF1 release upon gamma rotation, so ADP is not critically important for that. The discussion should be corrected accordingly.*

Thank you for the constructive comment. Along this, we have revised the manuscript.

- (11) *Lines 342-347 - I think that the data on alanine substitutions are interesting and informative; may be include them in the results section?*

Along this suggestion, we provide the additional data and information on alanine substitutions as Supplementary text and Supplementary Fig. 11.

- (12) *Finally, citing some recent reviews on the regulatory mechanisms of ATP synthase might also improve the manuscript.*

Along this comment, we added the recent review papers on regulation of ATP synthase (Cheuk A and Meier T, *Biochem. Soc. Trans.* 2021; Mendoza-Hoffmann F, *et al.*, *Microorganisms*. 2022) in adequate places in the revised manuscript (No. 29 & 30 with new numbering).

- *Additional comments shown in the attached pdf file*

- (13) *Line 44 – ‘Bacillus’ should be italic*

- (14) *Line 51 – Should add ‘is’ before ‘highly’*

- (15) *Line 79 – Remove the ‘it’ before F_1 (same as the comment (21)).*

- (16) *Line 337 – ‘requisite’ may be replaced with ‘required’*

We appreciate his/her careful reading of the manuscript. Along these comments, we have

revised the manuscript.

(17) Line 98 – What is ‘ φ ’?

‘ φ ’ is a diameter of the magnetic beads used in this work. To avoid misunderstanding, we added the words ‘beads diameter’ before ‘ φ ’ in the revised manuscript.

(18) Line 173 – ‘false’ may be replaced with ‘inability’ or ‘fail’

Along this comment, we have replaced ‘false’ with ‘failure’.

Reviewer #2

The manuscript demonstrates the mechanism of release of the inhibitor IF1 from ATP-synthase, using magnetic tweezers to mimic and assist ATP synthesis. Technically it is exemplary, the conclusions are demonstrated beautifully, clearly, and beyond doubt using a method that while not novel remains extremely challenging. I am not qualified to judge the significance of the result to the field of inhibitory control of ATP synthase. From the perspective of single-molecule biology, it is of great interest to a general reader as an example of the use of single molecule methods to reveal mechanistic details in biological molecular machines.

We sincerely appreciate the high evaluation by the Reviewer 2.

Reviewer #3

This study reports single-molecule manipulation experiments with immobilized bovine F1 linked to magnetic beads to elucidate the molecular mechanism of F1 inhibition by its natural inhibitory peptide IF1. The results showed that IF1-inhibited F1 was efficiently activated only when F1 was forcibly rotated (with magnetic tweezers) in the clockwise (ATP synthesis) direction, but not in the counter-clockwise direction (ATP hydrolysis), and the activation was enhanced in the presence of ADP and Pi. These observations suggest that the principal mechanism for the pmf-induced activation of IF1-inhibited FoF1 (in membranes) is the ejection of IF1 from F1 by pmf-driven Fo motor rotation. The results obtained with IF1 mutants strongly support a model in which the inhibitory effect of IF1 is mainly due the prevention of subunit beta conformational changes rather than changes in the interactions of IF1 with subunit gamma. These are major findings that provide novel and important insights into the mechanisms that regulate the activity of F1FO complexes. The presentation of the work is excellent.

We are most grateful for this reviewers' positive and encouraging comments on the work and are delighted to add the additional information requested.

● *Minor comment*

(19) In the introduction (lines 52, 53), the authors may want to include a previous reference (Lefebvre-Legendre et al. Mol. Microbiol 2003) showing that F1-mediated ATP hydrolysis is essential for cell viability, for instance in cells of Saccharomyces cerevisiae lacking functional mitochondrial DNA (and hence the FO) where this activity is absolutely needed to maintain a sufficient mitochondrial membrane potential by the ADP/ATP translocase. In these pure ATP hydrolysis conditions, inhibition of F1 by IF1 would be lethal rising the possibility that IF1 does not necessarily bind to the F1 when it cannot dock to FO?

Thank you for the careful reading and comments. The original manuscript attempted to mean that the hydrolysis activity of isolated F₁ before assembling onto F_o, but not ATP hydrolysis coupled with proton pumping for membrane voltage maintenance. To clarify this point, we revised the manuscript (line 55-56).

(20) The formulation "activation of IF1 inhibition" used all along the manuscript to indicate that F1 inhibition by IF1 is released is a bit confusing because it suggests instead that F1 becomes inhibited by IF1.

We appreciate the comment. To avoid misunderstandings, we define the meaning of "activation of/from IF₁ inhibition" as "IF₁ ejection from the catalytic site of F₁ and the recovery of catalysis in F₁" in the first place of the revised manuscript (line 84).

(21) line 79. Remove the 'it' before F1 (same as the comment (15)).

Along this comment, we removed the corresponding 'it'.